# The linear feedback precipitation model (LFPM 1.0) – a simple and efficient model for orographic precipitation in the context of landform evolution modeling

Stefan Hergarten[1] and Jörg Robl[2]

[1]Institut für Geo- und Umweltnaturwissenschaften, Albert-Ludwigs-Universität Freiburg, Freiburg, Germany
[2]Fachbereich Geographie und Geologie, Paris Lodron Universität Salzburg, Salzburg, Austria

**Correspondence:** Stefan Hergarten
(stefan.hergarten@geologie.uni-freiburg.de)

**Abstract.** The influence of climate on landform evolution has received great interest over the past decades. While many studies aim at determining erosion rates or parameters of erosion models, feedbacks between tectonics, climate and landform evolution have been discussed, but addressed quantitatively only in a few modeling studies. One of the problems in this field is that coupling a large-scale landform evolution model with a regional climate model would dramatically increase the theoretical and numerical complexity. Only a few simple models are available so far that allow a numerical efficient coupling between topography-controlled precipitation and erosion. This paper fills this gap by introducing a quite simple approach involving two vertically integrated moisture components (vapor and cloud water). The interaction between both components is linear and depends on altitude. This model structure is in principle the simplest approach that is able to predict both orographic precipitation at small scales and a large-scale decrease in precipitation over continental areas without introducing additional assumptions. Even in combination with transversal dispersion and elevation-dependent evapotranspiration, the model is of linear time complexity and increases the computing effort of efficient large-scale landform evolution models only moderately. Simple numerical experiments applying such a coupled landform evolution model show the strong impact of spatial precipitation gradients on mountain range geometry including steepness and peak elevation, position of the principal drainage divide, and drainage network properties.

## 1 Introduction

The redistribution of moisture from the oceans towards continental domains governs the global erosion engine. Spatial variability in precipitation and hence in the availability of water or ice as principal agents of erosion control the shape of landforms (e.g, Ellis et al., 1999; Willett, 1999; Anders et al., 2008; Bonnet, 2009; Menking et al., 2013; Colberg and Anders, 2014; Goren et al., 2014; Chen et al., 2019; Han et al., 2015; Paik and Kim, 2021). However, feedbacks between topography, precipitation and erosion may even make it difficult to distinguish between cause and effect (Molnar and England, 1990).

Long-term fluvial erosion is a field where simple numerical models have been applied with great success for some decades. The simplest model in this context is often referred to as the stream-power incision model (SPIM) and is the key component of

several models of long-term fluvial landform evolution (for an overview, see, e.g., Willgoose, 2005; Wobus et al., 2006). The SPIM considers rivers as linear elements (so without explicitly accounting for the width and the cross-sectional shape) and predicts the erosion rate $E$ as a function of the upstream catchment size $A$ and the channel slope $S$ in the form

$$E = K A^m S^n. \tag{1}$$

While the exponents $m$ and $n$ are kept constant, all site-specific influences on erosion are subsumed in a single lumped parameter $K$, called erodibility. The SPIM implements the concept of detachment-limited erosion (Howard, 1994) in the sense that all particles entrained by the river are immediately swept out of the system. The applicability of this concept even to bedrock rivers in high mountain regions has been questioned (e.g., Turowski, 2012). However, several extensions of the SPIM by sediment transport were proposed (e.g., Whipple and Tucker, 2002; Davy and Lague, 2009; Hergarten, 2020), where efficient numerical schemes have become available recently (Yuan et al., 2019; Hergarten, 2020). Even extensions towards glacial erosion were recently proposed (Deal and Prasicek, 2021; Hergarten, 2021a).

The SPIM and its derivates are well-suited for problems of tectonic geomorphology, e.g., variations in uplift rate or contrasts in lithology. In turn, the occurrence of the erodibility as a single, lumped parameter is a serious limitation concerning the influence of precipitation.

A framework for extending the SPIM towards a spatial variation in precipitation was already applied in several studies (e.g., Yanites and Ehlers, 2012; Goren et al., 2014; Garcia-Castellanos and Jiménez-Munt, 2015; Salles, 2016; Yuan et al., 2019). The idea is that erosion rates should rather depend on discharge than on catchment size, although the SPIM (Eq. 1) is typically written in terms of catchment size for historical reasons. Let $P$ be the effective precipitation for the moment, i.e., the part of the precipitation that contributes to discharge. If we assume that the actual erodibility $K$ refers to a given uniform reference precipitation $P_0$ and thus to a reference discharge $q_0 = P_0 A$, the catchment size $A$ can be replaced by $\frac{q_0}{P_0}$ in Eq. (1). It is then assumed that Eq. (1) holds for any discharge $q$ if $A$ is replaced by $\frac{q}{P_0}$. Since the equations become somewhat cumbersome if $q$ is replaced by the integral of $P$ over the upstream catchment, Hergarten (2021a) defined

$$A_{\text{eq}} = \frac{q}{P_0} \tag{2}$$

as the catchment-size equivalent of the discharge. It defines the catchment size needed to generate the actual discharge $q$ at the reference precipitation $P_0$. The advantage of using this terminology is that all relations in the context of erosion keep their simplicity, just with $A_{\text{eq}}$ instead of $A$.

One might think of using scenarios of a regional climate model for computing precipitation, e.g., the Weather Research and Forecasting (WRF) Model (Skamarock et al., 2021). The recent version of this model can in principle be run on PCs at spatial resolutions of a few kilometers, which allows for a consideration of orographic effects at the catchment scale. However, there would still be a huge imbalance between the complexity of the precipitation model and the simplicity of the erosion model. This imbalance would not only concern the computing effort, but also the level of detail of the prediction. While we might think of long-term mean precipitation rates in the extension of the SPIM by nonuniform precipitation described above, individual large rainstorms and related floods contribute much to landform evolution in reality. So even the question whether an ensemble

average or rather some kind of maximum over scenarios of a regional climate model yields a better input for landform evolution modeling is nontrivial.

Preserving the simplicity of the SPIM and its derivates requires simple models focusing on orographic effects on the relative precipitation $\frac{P}{P_0}$, which allows for computing $A_{eq}$ (Eq. 2). The main challenge is finding a level of complexity much below that of regional climate models that still provides new insights into landform evolution. On a qualitative level, reproducing an increased precipitation rate at the windward side of orogens and a rain shadow behind mountains would be some minimum requirement. Taking into account larger scales than the width of individual orogens, it may also be desirable to reproduce the overall decrease in precipitation with increasing distance from the reservoir of moisture (typically an ocean).

On a fundamental level, even extremely simple approaches have been used. Goren et al. (2014) distinguished between the windward side and the leeward side of a mountain belt just by the main drainage divide and assigned an increased relative precipitation to the windward region. This extremely simple model turned out to be sufficient for explaining a shift and an asymmetry in the drainage divide.

In turn, the models proposed by Roe et al. (2003), Smith and Barstad (2004), and Garcia-Castellanos (2007) use the concept of vertically integrated water contents and the respective fluxes per unit width. Assuming steady-state conditions, precipitation is derived from the negative divergence of the flux per unit width. All these models bring the topography into play by a thermodynamic equilibrium that depends on altitude via temperature.

The earliest among these models (Roe et al., 2003), however, does not model any fluxes explicitly, but directly proposes an equation for the divergence and thus for precipitation. The model predicts the rate of precipitation explicitly as a function of local surface elevation and slope in wind direction. As the only non-local component of the model, a Gaussian smoothing in the upwind direction was used in order to reduce effects of surface roughness. Due to these properties, the model is able to reproduce an increased precipitation at the windward side compared to the leeward side of a mountain belt, but fails to describe the large-scale shadow in a plane behind the mountain range or the decrease in precipitation with increasing distance to the ocean.

The two other models (Smith and Barstad, 2004; Garcia-Castellanos, 2007) consider spatially variable water contents and the respective fluxes, where transport at a given wind velocity is assumed. The model proposed by Smith and Barstad (2004) defines two components, interpreted as cloud water and hydrometeors. This model focuses on condensation and fallout at small scales, while it cannot predict transport over long distances (see Sect. 6). It therefore requires a refilling from an additional reservoir and is, similarly to the model of Roe et al. (2003), not able to predict large-scale precipitation patterns. In turn, the model of Garcia-Castellanos (2007) describes the vertically integrated water content by a single variable. Using a quite ingenious approach for describing deviations from equilibrium, it is able to capture the increase in precipitation with elevation as well the slow decrease in precipitation with increasing distance from the ocean. In turn, it requires an artificial smoothing at small scales, similarly to the model of Roe et al. (2003).

It seems that the model of Smith and Barstad (2004) (SB model in the following) received the biggest attention in the landform evolution modeling community among these models. It was adopted by some other authors in the context of co-evolution of topography and climate (e.g, Anders et al., 2008; Han et al., 2015; Paik and Kim, 2021), although the model

of Garcia-Castellanos (2007) has some advantages (see Sect. 6). The main advantage of the SB model seems to be that it can be implemented numerically on a regular grid using a forward and backward Fourier transform without the need to carry additional variables and to think about numerical stability and efficiency.

The goal of this study is developing a model that captures both the direct response of precipitation to changes in topography and large-scale precipitation patterns without the need for ad hoc assumptions such as an additional reservoir or smoothing. Beyond this, the numerical complexity should be not much higher than in the existing models. In particular, the linear time complexity (i.e., that the computing effort increases only linearly with the grid size) achieved by contemporary fluvial landform evolution models (Hergarten and Neugebauer, 2001; Braun and Willett, 2013; Yuan et al., 2019; Hergarten, 2020) should be preserved.

## 2 Model description

The model developed in the following is inspired by the concepts of Smith and Barstad (2004) and Garcia-Castellanos (2007). Similarly to these models, we describe the distribution of water in the atmosphere in terms of vertically integrated water contents measured in meters, which can be interpreted as water column heights. Following the ideas of Smith and Barstad (2004), we use two components, while the model of Garcia-Castellanos (2007) uses a single component and thus seems to be simpler at first sight. However, we will see in Sect. 4.1 that the effort of using two components pays off.

### 2.1 The governing equations

Let $Q_v$ be the content of vapor and $Q_c$ be the content of cloud water, both vertically integrated and measured as the height of a water column. Following the concepts of Smith and Barstad (2004) and Garcia-Castellanos (2007), we assume that advection with a given velocity is the predominant transport mechanism. If $v_{v/c}$ is the respective velocity of advection, the advective flux per unit width is

$$F_{v/c} = Q_{v/c} v_{v/c} \tag{3}$$

(measured in square meters per second), where the subscript v/c means that the relation holds for either vapor (v) or cloud water (c).

In a general formulation, $F_{v/c}$ and $v_{v/c}$ would be vectors in direction of advection. Let us, for simplicity, assume that the coordinate system is aligned in such a way that advection acts in x-direction. In addition, we assume dispersion in y-direction, i.e., in direction normal to the advection. Then the vertically integrated moisture balance for each of the components reads

$$\frac{\partial Q_{v/c}}{\partial t} = -\frac{\partial}{\partial x} F_{v/c} + \frac{\partial}{\partial y} \left( L_d v_{v/c} \frac{\partial Q_{v/c}}{\partial y} \right) + S_{v/c}, \tag{4}$$

where $S_{v/c}$ is a source term (measured in meters per second) describing the interaction between the components and the loss by precipitation.

The dispersion term used in Eq. (4) is a specific form of a diffusion term with a diffusivity $D = L_d v_{v/c}$, where $L_d$ is the dispersion length. Dispersion terms in advection equations typically arise from a spatial variability in velocity that is not

resolved by the large-scale description of the flow field. Assuming a constant dispersion length $L_d$ reflects the idea that the fluctuations in velocity and thus the diffusivity are directly proportional to the large-scale velocity. However, assuming a constant dispersion length is not essential for the model developed here. Similarly to assuming the same dispersion length for both components, it is just a convenient choice that keeps the equations simple.

Dispersion in longitudinal direction is not taken into account, although there is no reason why it should be smaller than transversal direction. The reason for including only transversal dispersion is that it has a larger effect on the properties of the model. Transversal dispersion is the only process that links points with different y-values. Without transversal dispersion, the precipitation pattern would fall into a set of individual lines parallel to the x-axis. So transversal dispersion is an essential component of the approach in combination with two-dimensional landform evolution models. In turn, we will see in Sect. 4.1 that longitudinal dispersion is not essential for the properties of the model, while it would make the numerical treatment more complicated (Sect. 3).

Since the time scales of processes in the atmosphere are much shorter than the time scales involved in landform evolution, steady-state conditions can be assumed in Eq. (4). If we furthermore assume that the velocities are constant, Eq. (4) can be written conveniently in terms of the fluxes per unit width (Eq. 3):

$$-\frac{\partial F_{v/c}}{\partial x} + L_d \frac{\partial^2 F_{v/c}}{\partial y^2} + S_{v/c} = 0. \tag{5}$$

Following the concepts of Smith and Barstad (2004), we assume that condensation (from $Q_v$ to $Q_c$) and precipitation (from $Q_c$) are linear processes with given time constants $\tau_c$ and $\tau_f$, respectively. In contrast to this model and also to the model of Garcia-Castellanos (2007), we do not introduce an equilibrium water content explicitly. Instead, we start from a more fundamental level by considering condensation of vapor and re-evaporation of cloud water (e.g., Roe, 2005) as competing processes in the form

$$S_v = -\frac{Q_v - \alpha Q_c}{\tau_c}. \tag{6}$$

The nondimensional coefficient $\alpha$ defines the dynamic equilibrium between the two processes. An equilibrium between $Q_c$ and $Q_v$ is achieved if $\frac{Q_v}{Q_c} = \alpha$. Rewriting Eq. (6) in terms of fluxes per unit width yields

$$S_v = -\frac{F_v - \beta F_c}{L_c}, \tag{7}$$

with the length scale of condensation $L_c = v_v \tau_c$ and the modified coefficient $\beta = \frac{v_v}{v_c} \alpha$ (still nondimensional).

Since the extension of the conversion of vapor into cloud water by a (negative) linear feedback term (evaporation) is the key idea behind our approach, the model is called linear feedback precipitation model (LFPM) in the following.

The rate of precipitation (measured in meters per second) can also be expressed in terms of the flux per unit width according to

$$P = \frac{Q_c}{\tau_f} = \frac{F_c}{L_f}, \tag{8}$$

with the length scale $L_{\text{f}} = v_{\text{c}}\tau_{\text{f}}$. Then the source term of $Q_{\text{c}}$ is

$$S_{\text{c}} = -S_{\text{v}} - P = \frac{F_{\text{v}} - \beta F_{\text{c}}}{L_{\text{c}}} - \frac{F_{\text{c}}}{L_{\text{f}}}, \tag{9}$$

and the full system of differential equations for the two fluxes reads

$$-\frac{\partial F_{\text{v}}}{\partial x} + L_{\text{d}}\frac{\partial^2 F_{\text{v}}}{\partial y^2} - \frac{F_{\text{v}} - \beta F_{\text{c}}}{L_{\text{c}}} \qquad\qquad = 0 \tag{10}$$

$$-\frac{\partial F_{\text{c}}}{\partial x} + L_{\text{d}}\frac{\partial^2 F_{\text{c}}}{\partial y^2} + \frac{F_{\text{v}} - \beta F_{\text{c}}}{L_{\text{c}}} \qquad -\frac{F_{\text{c}}}{L_{\text{f}}} = 0. \tag{11}$$

## 2.2 The effect of topography

Orographic precipitation is related to a dependence of the equilibrium on altitude (e.g., Roe, 2005). Since the re-evaporation of cloud water requires energy, altitude has an immediate effect here. The rate of re-evaporation should decrease with decreasing temperature and thus with increasing altitude. As the simplest approach, we consider only this effect and assume that the length scales of condensation ($L_{\text{c}}$) and fallout ($L_{\text{f}}$) are constant. The Arrhenius relation

$$\beta \propto e^{-\frac{a}{T}} \tag{12}$$

with a constant $a$ provides the simplest model for the dependence of $\beta$ on the temperature $T$, where both $a$ and $T$ are measured in Kelvin. Using a linear decrease of temperature with altitude $H$,

$$T = T_0 - \Gamma H, \tag{13}$$

where $T_0$ is the temperature at sea level and $\Gamma$ the lapse rate (measured in Kelvin per meter), Eq. (12) can be written in the form

$$\frac{\beta}{\beta_0} = \frac{e^{-\frac{a}{T_0 - \Gamma H}}}{e^{-\frac{a}{T_0}}} = e^{-\left(\frac{a}{T_0 - \Gamma H} - \frac{a}{T_0}\right)} = e^{-\frac{a\Gamma H}{T_0(T_0 - \Gamma H)}}, \tag{14}$$

where $\beta$ refers to the altitude $H$ and $\beta_0$ to sea level. Defining $H_0 = \frac{T_0^2}{a\Gamma}$, Eq. (14) can be written in the form

$$\beta = \beta_0 e^{-\frac{H}{H_0\left(1 - \frac{\Gamma H}{T_0}\right)}}. \tag{15}$$

For simplicity, we assume that Eq. (15) also holds for the vertically integrated cloud water content with $H$ as the surface elevation and neglect the term $\frac{\Gamma H}{T_0}$. The latter is a first-order approximation concerning $H$, which requires that the decrease in temperature $\Gamma H$ is small compared to the absolute temperature $T_0$ at sea level. Using these approximations, Eq. (15) reduces to

$$\beta = \beta_0 e^{-\frac{H}{H_0}}. \tag{16}$$

This relation describes the decrease of $\beta$ by a single lumped parameter $H_0$, which defines a vertical length scale and describes the elevation where $\beta$ has decreased by a factor $e$ compared to sea level. While the description of the height-dependence by a single, lumped parameter is convenient, it is not an essential part of the LFPM. Any other relation, e.g., the more elaborate version used by Garcia-Castellanos (2007), which does not rely on the two approximations introduced above, could be used as well.

## 2.3 Boundary conditions

Since the system of differential equations defined by Eqs. (10) and (11) is of first order in $x$ and of second order in $y$, it is a parabolic system. Finding a unique solution in a rectangular domain ($0 \leq x \leq x_{\max}$, $0 \leq y \leq y_{\max}$) requires boundary conditions at $x = 0$, $y = 0$, and $y = y_{\max}$ (but not at $x = x_{\max}$).

Since moisture is coming in at $x = 0$, it is straightforward to define $F_v$ and $F_c$ there. Then the integral of the total influx $F = F_v + F_c$ over this boundary defines the total amount of water available for precipitation in the domain. However, the question how to distribute a given total influx $F$ to $F_v$ and $F_c$ is not trivial and requires more knowledge about the properties of the model. It will be addressed in Sect. 4.1.

All types of boundary conditions could be used at $y = 0$ and $y = y_{\max}$. Neumann boundary conditions or periodic boundary
conditions are more useful than Dirichlet boundary conditions here since the $F_v$ and $F_c$ are fluxes along these boundaries, and it is not trivial to define reasonable prescribed values for $F_v$ and $F_c$. Homogeneous Neumann boundary conditions define $\frac{\partial F_{v/c}}{\partial y} = 0$, which means that there is no transversal dispersion across these boundaries. The implementation in the landform evolution OpenLEM presented in the following section uses periodic boundary conditions in y-direction by default, which are convenient in many applications.

## 3  Numerical implementation

Taking into account advection only along one of the coordinate axes and neglecting longitudinal dispersion considerably facilitates the numerical implementation of the model. Let us first rewrite Eqs. (10) and (11) in matrix form

$$-\frac{\partial \boldsymbol{F}}{\partial x} + L_d \frac{\partial^2 \boldsymbol{F}}{\partial y^2} - \frac{1}{L_c} \mathbf{A} \boldsymbol{F} = 0, \tag{17}$$

where

$$\boldsymbol{F} = \begin{pmatrix} F_v \\ F_c \end{pmatrix}, \quad \mathbf{A} = \begin{pmatrix} 1 & -\beta \\ -1 & \beta + \phi \end{pmatrix}, \quad \text{and} \quad \phi = \frac{L_c}{L_f}. \tag{18}$$

Let us further assume unit grid spacing in both directions, $\delta x = \delta y = 1$, for simplicity. This means that the length scales $L_d$, $L_c$, and $L_f$ must be measured in terms of the grid spacing in the following. If we use a left-hand (so upwind) difference quotient for the advection term, the discretized form of Eq. (17) can be written in the form

$$-\boldsymbol{F}_{x,y} + L_d \left( \boldsymbol{F}_{x,y-1} - 2\boldsymbol{F}_{x,y} + \boldsymbol{F}_{x,y+1} \right) - \frac{1}{L_c} \mathbf{A} \boldsymbol{F}_{x,y} = -\boldsymbol{F}_{x-1,y}, \tag{19}$$

where the indices $x$ and $y$ correspond to the positions. So the values $\boldsymbol{F}_{x,\cdot}$ can be computed from the values $\boldsymbol{F}_{x-1,\cdot}$ by solving a one-dimensional problem (in y-direction). The respective linear equation has a tridiagonal structure of $2 \times 2$ blocks and can be written in the form

$$-L_d \boldsymbol{F}_{x,y-1} + \mathbf{D} \boldsymbol{F}_{x,y} - L_d \boldsymbol{F}_{x,y+1} = \boldsymbol{F}_{x-1,y} \tag{20}$$

with the $2 \times 2$-matrix

$$\mathbf{D} = (1 + 2L_\mathrm{d}) \, \mathbf{1} + \frac{1}{L_\mathrm{c}} \mathbf{A} \tag{21}$$

and the $2 \times 2$ identity matrix $\mathbf{1}$. This equation system can be solved, e.g., by the direct Gaussian scheme based on $2 \times 2$ blocks.

The examples shown in the following section are computed using the open-source landform evolution model OpenLEM. This model already contains up-to-date implementations of fluvial erosion such as the shared stream-power model (Hergarten, 2020), which will be used in Sect. 8. All components of OpenLEM are of linear time complexity at arbitrary time step lengths, which means that the numerical effort increases only linearly with the size of the lattice. The computation of the precipitation proposed here preserves this property. Independently of the size of the lattice, we found an increase in computing time by a factor of about 2.4 compared to the simplest form of the SPIM and a factor of about 2.2 compared to the shared stream-power model, which includes sediment transport. This increase is owing to taking into account transversal dispersion, where Neumann boundary conditions would be cheaper than the periodic boundary conditions used in OpenLEM.

## 4 Fundamental properties of the model

### 4.1 Characteristic length scales

Let us for the moment consider the model only in longitudinal direction, i.e., without the dispersion term, and let us assume a constant elevation for the moment. Then the set of parameters consists of two horizontal length scales $L_\mathrm{c}$ and $L_\mathrm{f}$ and a nondimensional parameter $\beta$. In this section, it is shown that the relevant length scales that characterize the properties of the model differ from $L_\mathrm{c}$ and $L_\mathrm{f}$.

In this situation, Eq. (17) reduces to a linear system of two ordinary differential equations,

$$\frac{\partial \boldsymbol{F}}{\partial x} = -\frac{1}{L_\mathrm{c}} \mathbf{A} \boldsymbol{F}. \tag{22}$$

The behavior of the solutions are determined by the eigenvalues of the matrix $\mathbf{A}$ defined in Eq. (18). These are found by solving the characteristic equation of $\mathbf{A}$,

$$\lambda^2 - (1 + \beta + \phi) \, \lambda + \phi = 0, \tag{23}$$

which yields

$$\lambda_\pm = \frac{1 + \beta + \phi}{2} \pm \sqrt{\left( \frac{1 + \beta + \phi}{2} \right)^2 - \phi}. \tag{24}$$

Since $\beta$ and $\phi$ are nondimensional, the eigenvalues $\lambda_\pm$ are also nondimensional. The eigenvalues describe exponentially decaying solutions of the form $e^{-\frac{\lambda_\pm}{L_\mathrm{c}} x}$. These solutions can also be written in the form $e^{-\frac{x}{L_\mathrm{l}}}$ and $e^{-\frac{x}{L_\mathrm{s}}}$, respectively, where

$$L_\mathrm{l} = \frac{L_\mathrm{c}}{\lambda_-} \quad \text{and} \quad L_\mathrm{s} = \frac{L_\mathrm{c}}{\lambda_+} \tag{25}$$

describe the respective length scales of the decay. Since $\lambda_- \leq 1$ and $\lambda_- \leq \phi$ for all values of $\beta$ and $\phi$,

$$L_\mathrm{l} > \max\{L_\mathrm{c}, L_\mathrm{f}\} \tag{26}$$

for $\beta > 0$. In turn, $\lambda_- \lambda_+ = \phi$, and thus

$$L_\mathrm{l} L_\mathrm{s} = L_\mathrm{c} L_\mathrm{f}. \tag{27}$$

As a consequence,

$$L_\mathrm{s} < \min\{L_\mathrm{c}, L_\mathrm{f}\} \tag{28}$$

for $\beta > 0$. So the approach based on the dynamic equilibrium creates two characteristic length scales outside the range between $L_\mathrm{c}$ and $L_\mathrm{f}$. These scales differ even if we assume $L_\mathrm{c} = L_\mathrm{f}$ as suggested by Smith and Barstad (2004).

The longer length scale $L_\mathrm{l}$ describes the ability to transport moisture over large distances. Smith and Barstad (2004) suggested time scales of 200 s to 2000 s for the conversion of cloud water and for fallout, corresponding to length scales $L_\mathrm{c}$ and $L_\mathrm{f}$ of 10 km to 100 km at wind speeds of 50 ms$^{-1}$. If we, e.g., assume $\beta = 10$, $L_\mathrm{l}$ is in the range between 119 km and 1191 km. So the transport range may be considerably larger than the length scales $L_\mathrm{c}$ and $L_\mathrm{f}$ of the involved processes. This property is essential for simulating long-range transport over large continental areas with a closed water balance.

Interestingly, Makarieva et al. (2009) indeed found an exponential decay of mean precipitation rates with increasing distance from the ocean at nonforested areas in a worldwide analysis. They found a decay length of about 600 km, which is 6 to 60 times larger than the reasonable range of $L_\mathrm{c}$ and $L_\mathrm{f}$, which even refers to quite high wind speeds. This result supports the idea behind the LFPM and provides an idea about the order of magnitude of $L_\mathrm{l}$.

The mode of long-range transport is obtained by inserting $\lambda_-$ into the first row of the eigenvalue equation $\mathbf{A}\boldsymbol{F} = \lambda \boldsymbol{F}$,

$$F_\mathrm{v} - \beta F_\mathrm{c} = \lambda_- F_\mathrm{v}, \tag{29}$$

and thus

$$\frac{F_\mathrm{c}}{F_\mathrm{v}} = \frac{1 - \lambda_-}{\beta} = \frac{1 - \frac{L_\mathrm{c}}{L_\mathrm{l}}}{\beta} = \frac{1}{\frac{L_\mathrm{l}}{L_\mathrm{f}} - 1}. \tag{30}$$

According to Eq. (7), this ratio is $\frac{F_\mathrm{c}}{F_\mathrm{v}} = \frac{1}{\beta}$ in equilibrium. So the vapor content is slightly above its equilibrium value in the long-range transport mode, which results in a low net rate of condensation.

Figure 1 illustrates the long-range transport for a boxcar-shaped topography with a height $H = H_0$. All properties are considered as nondimensional values. The parameter values are $L_\mathrm{c} = L_\mathrm{f} = 1$, and $\beta_0 = 10$. The incoming fluxes are $F_\mathrm{v} = 10$ and $F_\mathrm{c} = 0$ at $x = 0$. According to Eqs. (24) and (26), the length scale of long-range transport is $L_\mathrm{l} \approx 11.9$ at sea level ($H = 0$). Both fluxes and the rate of precipitation decrease exponentially with this length scale for $x < 5$ and $x > 10$, except for the beginning of the ranges.

The plateau ($H = H_0$) is characterized by a lower value of $\beta = e^{-1} \approx 0.37$ according to Eq. (7), resulting in a lower length scale $L_\mathrm{l} \approx 5.5$. So the precipitation is higher at the plateau, but in turn decreases more rapidly with $x$. This difference is reflected in a lower ratio $\frac{F_\mathrm{c}}{F_\mathrm{v}}$, so in a lower ability to keep moisture in form of vapor.

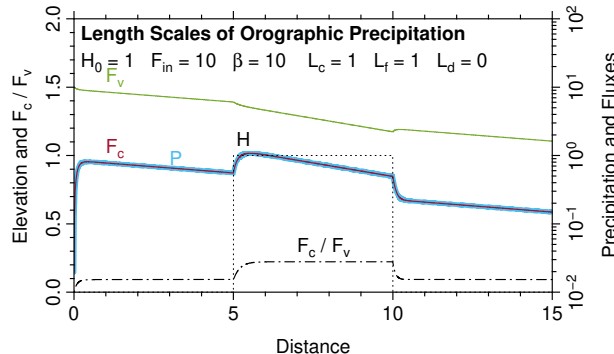

**Figure 1.** Principal properties of the model in 1D for a boxcar-shaped topography. Fluxes and precipitation rates are shown on logarithmic axes in order to emphasize the exponential decrease.

In turn, the short length scale $L_s$ describes the adjustment if the ratio of $F_c$ and $F_v$ deviates from Eq. (30). Since these deviations predominantly arise from changes in topography (via the elevation-dependence of $\beta$), the length scale $L_s$ can be considered as the length scale of orographic precipitation.

Three transition zones characterized by $L_s$ occur in Fig. 1, starting at $x = 0$, $x = 5$, and $x = 10$, respectively. The length scale of the first and the third transition ($H = 0$) is $L_s \approx 0.08$, while it is $L_s \approx 0.18$ for the second transition ($H = H_0$). In general, $L_s$ increases with elevation, while the length scale of long-range transport $L_l$ decreases with elevation. Their product is constant according to Eq. (27).

While the second and the third transition arise from changes in topography, the first transition occurs because it is assumed that the influx only contains vapor ($F_v = 10$), but no cloud water ($F_c = 0$), which is far off from equilibrium and from the long-range transport mode. It is therefore useful to adjust the boundary condition in such a way that the incoming fluxes are in the long-range transport mode described by Eq. (30). Then the incoming flux of cloud water must be

$$F_c = F\frac{L_f}{L_l}, \tag{31}$$

where $F$ is the total incoming flux, and $F_v = F - F_c$. This modified boundary condition is used in all subsequent examples throughout this study.

The non-instantaneous reaction to abrupt changes in topography is a central property of the model. Without this, the small-scale roughness of topography would directly affect the precipitation pattern, so that an additional smoothing procedure would be required. Longitudinal dispersion could also used for smoothing, but as the LFPM generates a scale of smoothing on its own, taking into account longitudinal dispersion is not urgently required. Taking this result into account, our approach based on two components of water content with a two-way conversion is some kind of minimum model that is able to capture both continentality (a slow decrease in precipitation at large scales) and a delayed reaction to small-scale changes in topography.

While the model in its original form involves two longitudinal length scales $L_c$ and $L_f$, a transversal length scale $L_d$, a vertical length scale $H_0$, and a nondimensional parameter $\beta_0$ (referring to sea level), it is also possible to replace $\beta_0$ by the

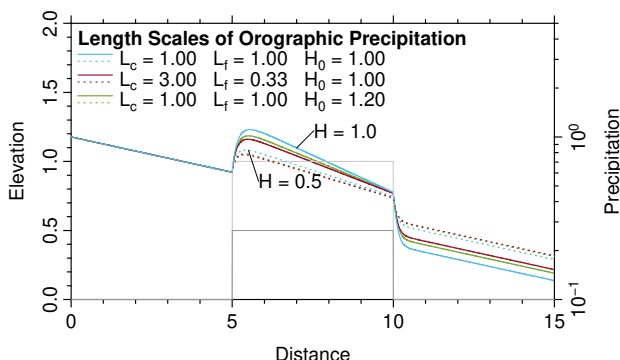

**Figure 2.** Effect of $L_c$ and $L_f$, where $L_l$ (at sea level) and the product $L_c L_f$ were kept constant. The solid lines refer to the original scenario ($H = 1$) and the dashed lines to a reduced topography ($H = 0.5$). The results for $L_c = \frac{1}{3}$ and $L_f = 3$ are practically the same as for $L_c = 3$ and $L_f = \frac{1}{3}$.

length scale of long-range transport $L_l$ (alternatively also by $L_s$, but that would be less useful). The value of $\beta$ can be computed conveniently from Eq. (23):

$$\beta = \frac{\lambda^2 + \phi}{\lambda} - \phi - 1 \tag{32}$$

$$= \left(1 - \frac{L_c}{L_l}\right)\left(\frac{L_l}{L_f} - 1\right). \tag{33}$$

While this relation is valid for the respective values of $L_l$ and $\beta$ at any elevation, it is particularly useful for computing $\beta_0$ from $L_c$, $L_f$, and $L_l$ at sea level.

If we use $L_l$ instead of $\beta_0$, Eq. (27) reveals that the short length scale $L_s$ only depends on $L_l$ and on the product $L_c L_f$, while the individual values of $L_c$ and $L_f$ are not relevant for $L_s$ at sea level. This is, however, not the case for the elevation-dependence. Figure 2 illustrates the relevance of the individual values of $L_c$ and $L_f$ in combination with topography. The default scenario (solid blue line) is the same as in Fig. 1, except that the fluxes at the boundary were adjusted according to Eq. (31). In all scenarios, $L_l$ (at sea level) was kept constant (so not $\beta_0$). As expected, the behavior at sea level remains the same for $L_c \neq L_f$ (red lines) as long as the product $L_c L_f$ is constant. This is, however, not true at $H > 0$, where the increase in precipitation with elevation becomes distinctly weaker for $L_c \neq L_f$ (red vs. blue lines), regardless which of the values is greater.

The decrease in $L_l$ with elevation and thus the respective increase in precipitation can be computed from Eq. (16) according to

$$\frac{dL_l}{dH} = \frac{d\beta}{dH}\frac{dL_l}{d\beta} = \frac{\beta}{H_0}\frac{L_c}{\lambda_-^2}\frac{d\lambda_-}{d\beta}. \tag{34}$$

The remaining derivative can be computed from Eq. (23),

$$\frac{d\lambda_-}{d\beta} = \frac{\lambda_-}{2\lambda_- - (1 + \beta + \phi)} = -\frac{\lambda_-}{\lambda_+ - \lambda_-}. \tag{35}$$

Inserting this result into Eq. (34) yields

$$\frac{dL_l}{dH} = -\frac{\beta}{H_0}\frac{L_f L_l}{L_l - L_s} \tag{36}$$

after some basic transformations, and finally after inserting Eq. (33)

$$\frac{dL_l}{dH} = -\frac{1}{H_0}\frac{(L_l - L_c)(L_l - L_f)}{L_l - L_s}. \tag{37}$$

It is easily recognized that the second factor is always lower than $L_l$. So the decrease in $L_l$ and thus the increase in precipitation with elevation is always smaller than the decrease in $\beta$, which is characterized by the vertical scale $H_0$. The relation is symmetric concerning $L_c$ and $L_f$, and the elevation-dependence is strongest for $L_c = L_f = \sqrt{L_l L_s}$.

At least for topographies with moderate relief (in relation to $H_0$), a difference between $L_c$ and $L_f$ can be replaced by an increased value of $H_0$. If we define

$$\tilde{L}_c = \tilde{L}_f = \sqrt{L_c L_f} \quad \text{and} \quad \tilde{H}_0 = \frac{(L_l - \tilde{L}_c)^2}{(L_l - L_c)(L_l - L_f)}H_0, \tag{38}$$

the behavior of the model essentially remains the same for moderate elevations. This result is illustrated by the green lines in Fig. 2. The precipitation obtained for $L_c = L_f = 1$ with an increased reference elevation $H_0 = 1.2$ (Eq. 38) are close to those for $L_c = 3$ and $L_f = \frac{1}{3}$ with $H_0 = 1$ for the topography with $H = \frac{1}{2}$. For the higher topography with $H = 1$, however, the remaining deviation is larger.

Keeping in mind that the definition of $H_0$ in Sect. 2.2 already required some approximations, it is not a problem that $H_0$ has to be increased artificially if we replace different values of $L_c$ and $L_f$ by the same value $\sqrt{L_c L_f}$. If we accept that there is a residual overestimation of the effect of topography that increases with elevation, we can assume $L_c = L_f$ without losing much of the model's fundamental capabilities.

## 4.2 The influence of transversal dispersion

As mentioned above, transversal dispersion is the only component that prevents the model from falling into a set of independent one-dimensional models. In contrast to the other length scales of the model, however, the dispersion length cannot be interpreted directly as a spatial scale. It rather links longitudinal and transversal length scales of the moisture pattern.

Let us for the moment assume that condensation and fallout are switched off ($L_c = L_f \to \infty$) and that the topography is flat ($H = 0$). Let us further assume that the incoming flux (at $x = 0$) has some transversal variation in water content (either in $F_v$ or $F_c$ or in both) according to

$$\delta F(0, y) \propto \sin\left(\frac{\pi y}{L_y}\right), \tag{39}$$

where $L_y$ defines the length scale of this variation (half of the wavelength). Then Eq. (10) (or alternatively Eq. 11) yields

$$\delta F(x, y) \propto \delta F(0, y)e^{-\frac{x}{L_x}}, \tag{40}$$

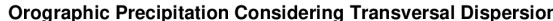

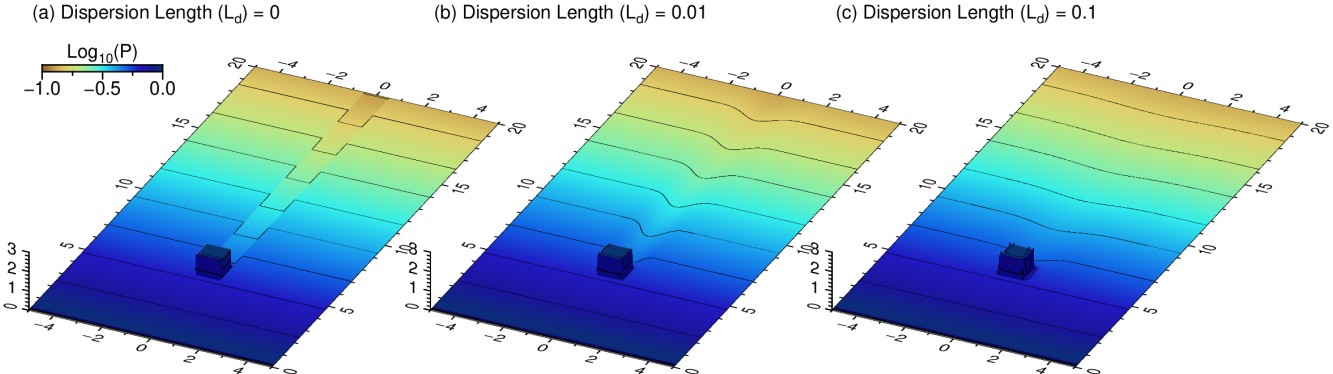

**Figure 3.** Effect of transversal dispersion on the precipitation behind an obstacle.

with

$$L_{\mathrm{x}} = \frac{L_{\mathrm{y}}^2}{\pi^2 L_{\mathrm{d}}}. \tag{41}$$

So small-scale transversal patterns decay much faster in direction of advection than large-scale patterns. The length scale of the decay, $L_{\mathrm{x}}$, decreases quadratically with the scale $L_{\mathrm{y}}$ of the transversal pattern. The length scale of dispersion, $L_{\mathrm{d}}$, describes

the strength of dispersion, so that an increase in $L_{\mathrm{d}}$ reduces $L_{\mathrm{x}}$.

Figure 3 illustrates the effect of dispersion for an obstacle of a width $L_{\mathrm{y}} = 1$ and a height $H = 1$, where the parameter values are the same as in the previous examples. Without transversal dispersion ($L_{\mathrm{d}} = 0$), the higher precipitation falling on the obstacle causes an infinite precipitation shadow. For $L_{\mathrm{d}} = 0.01$, the longitudinal scale of decay is $L_{\mathrm{x}} \approx 10$ (Eq. 41). The precipitation shadow has become considerable weaker at this distance behind the obstacle, but is still visible. Finally, the scale

of decay decreases to $L_{\mathrm{x}} \approx 1$ for $L_{\mathrm{d}} = 0.1$, so that the shadow vanishes rapidly behind the obstacle.

## 5   Extension by evapotranspiration

Evaporation including the transpiration by plants, called evapotranspiration, plays a major part in the water balance. While the potential rate of evapotranspiration mainly depends on the climatic conditions and on vegetation, the actual rate is often much lower due to limited availability of water at the surface and in the shallow subsurface.

However, the concept for including variations in precipitation in large-scale landform evolution models is not able to predict the availability of water. The water balance is stated in terms of fluxes, which are not directly related to amounts of stored water. Estimating the amount of stored water would require a model for the flow velocity and would introduce additional complexity. Garcia-Castellanos (2007) presented a first step in this direction by distinguishing lake areas and assigning a rate of evaporation to these areas.

Here we propose a simpler idea by assuming that the rate of evapotranspiration is proportional to the rate of precipitation instead of the amount of stored water. This means that a given fraction $\epsilon$ of the precipitation evaporates immediately. This leads to one additional term in Eq. (10), so that the system of differential equations turns into

$$-\frac{\partial F_v}{\partial x} + L_d\frac{\partial^2 F_v}{\partial y^2} - \frac{F_v - \beta F_c}{L_c} + \quad \epsilon \quad \frac{F_c}{L_f} = 0, \tag{42}$$

$$-\frac{\partial F_c}{\partial x} + L_d\frac{\partial^2 F_c}{\partial y^2} + \frac{F_v - \beta F_c}{L_c} - \quad \frac{F_c}{L_f} = 0. \tag{43}$$

While the total precipitation is still $P = \frac{F_c}{L_f}$, the effective precipitation that contributes to runoff is

$$P_{\text{eff}} = (1 - \epsilon)P = (1 - \epsilon)\frac{F_c}{L_f} \tag{44}$$

then.

In order to understand the effect of evapotranspiration, Eqs. (42) and (43) can be brought to the same form as Eqs. (10) and (11) by introducing an increased coefficient for re-evaporation in the atmosphere

$$\tilde{\beta} = \beta + \epsilon\frac{L_c}{L_f} \tag{45}$$

and an increased fallout length

$$\tilde{L}_f = \frac{L_f}{1 - \epsilon}. \tag{46}$$

Then $\phi$ (Eq. 18) changes to

$$\tilde{\phi} = (1 - \epsilon)\,\phi. \tag{47}$$

It is easily recognized that $\tilde{\beta} + \tilde{\phi} = \beta + \phi$. So only the last term $\phi$ in Eq. (24) is affected by evapotranspiration. The smaller eigenvalue $\lambda_-$ comes closer to zero then, while the greater eigenvalue $\lambda_+$ does not change much. So $L_1$ increases considerably, while $L_s$ remains almost constant, and thus

$$\tilde{L}_l \approx \frac{L_l}{1 - \epsilon} \tag{48}$$

according to Eq. (27).

While these results suggest that the effect of evapotranspiration could be mimicked by modifying the parameters $\beta$ and $L_f$ (Eqs. 45 and 46), it must be kept in mind that this only holds at constant elevation. Otherwise, $\beta$ depends on $H$, and thus also the effect of $\epsilon$ on $L_l$. Since $\beta$ describes the re-evaporation of water in the atmosphere, it makes sense to assume that the rate of evapotranspiration has the same dependence on $H$ as $\beta$,

$$\epsilon = \epsilon_0 e^{-\frac{H}{H_0}}, \tag{49}$$

although $\epsilon$ refers to the surface and $\beta$ to vertically integrated properties.

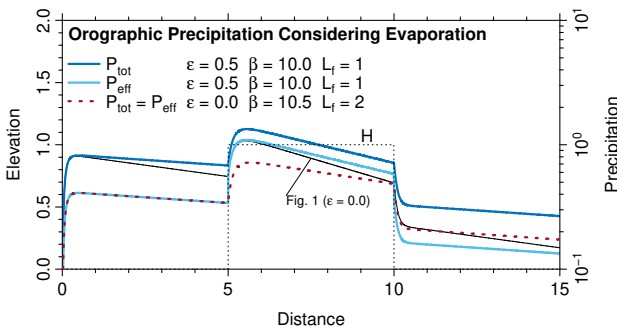

**Figure 4.** Effect of elevation-dependent evaporation on precipitation.

Then the dependence of Eq. (45) remains valid for all elevations. However, as $\epsilon$ decreases with elevation, the increase in the effective $L_f$ (Eq. 46) and thus also in the effective $L_l$ becomes weaker at large elevations. So mimicking evapotranspiration by adjusting $\beta$ and $L_f$ (Eqs. 45 and 46) would overestimate the effect of evaporation at large elevations or, in turn, underestimate the effect of elevation. Figure 4 illustrates this result for the example from Fig. 1, where an evaporation ratio $\epsilon_0 = 0.5$ is
385 assumed at sea level. The results are compared to the model without evaporation, but with modified parameter values $\tilde{\beta} = 10.5$ (Eq. 45) and $\tilde{L}_f = 2$ (Eq. 46). The effective precipitation in the mountain region and the effective length scale of transport differ by almost a factor of two for $H = H_0$. So mimicking evapotranspiration by adjusted parameters is only useful for small topography.

## 6 Comparison to existing models

As discussed in Sect. 1, the models of Smith and Barstad (2004) (SB model) and Garcia-Castellanos (2007) also use vertically integrated water contents and advective transport at a given wind velocity. In its spirit, the LFPM is somewhat similar to these models.

It may seem at first that taking into account transversal dispersion was the major progress of our approach. However, the numerical scheme proposed in Sect. 3 could in principle also be used for including transversal dispersion in the two other
models. The fundamental differences are hidden in details of the model structure that have a bigger effect than it seems at first.

The SB model assumes only a one-way coupling between the two moisture components. In our terminology, this would be $\beta = 0$ in Eqs. (10) and (11). Following the considerations of Sect. 4.1, the length scale of long-range transport of moisture is $L_l = \max\{L_c, L_f\}$ then, which is between about 10 km and 100 km at rather high wind speeds of 50 ms$^{-1}$. So assuming a one-way conversion practically removes the ability to transport moisture over large distances of several hundred kilometers.
Therefore, the SB model requires a permanent refilling of the water storages at some given background rate. When considering a large plain, the precipitation rate will always approach this prescribed background rate, regardless of the topography in front of the plain. So this model focuses on the behavior at intermediate scales, but cannot capture large-scale precipitation patterns.

This is presumably the reason why the two moisture components are interpreted as cloud water and hydrometeors in the SB model instead of vapor and cloud water in the LFPM.

Beyond this, the feedback parameter $\beta$ carries the information about the elevation-dependence in the LFPM. So the effect of topography must be included in another way if this feedback is not taken into account. Smith and Barstad (2004) assumed that the rate of conversion is not proportional to the absolute value of the cloud water content ($Q_v$ in the LFPM), but to the difference of this content towards an elevation-dependent equilibrium content. This results in an additional source or sink term in the equations, which carries all information about the topography.

In the earliest version of the SB model (Smith, 2003), it was assumed that the source term is directly proportional to the topographic slope in flow direction, $\frac{\partial H}{\partial x}$. So upslope flow introduces a positive source term, which is converted into precipitation with some lag and smoothing. Smith and Barstad (2004) proposed a more elaborate source term taking into account airflow dynamics in more detail.

In the following, we compare the two versions of the SB model to the LFPM in a one-dimensional example that describes the
rise to a plateau. In contrast to the similar example considered by Smith and Barstad (2004, Sect. 3c), we do not use a smooth arctangent function, but a ramp with a constant slope between two horizontal planes for clarity. Four different topographies are considered in Fig. 5, where nondimensional coordinates are used. Similarly to Fig. 1, $L_c = L_f$ is used as the horizontal length scale, while $H_0$ defines the vertical length scale. The four scenarios refer to plateau elevations of $H = H_0$ and $H = 4H_0$ and to ramp lengths of $2.5L_c$ and $10L_c$. All precipitation values are arbitrarily scaled, but by constant factors for each model
throughout all scenarios. So the absolute precipitation values cannot be compared among the SB model and the LFPM, but among different scenarios for the same model. The background rate is zero in all results of the SB model, so that the values shown in Fig. 5 should not be interpreted as absolute values, but as differences towards a prescribed background precipitation rate for the SB model.

The upslope version of the SB model shows the simplest behavior. The precipitation rate is zero (or equal to the prescribed
background rate) in front of the ramp as well as at the plateau far behind the ramp. Since the ramp introduces a constant source term, a constant precipitation rate is also approached at the ramp if the ramp is sufficiently long (Fig. 5b,d). Otherwise, the increase in precipitation ceases at the end of the ramp before a constant rate is approached, which results in a distinct maximum in precipitation at the transition to the plateau.

For the version of the SB model with the more elaborate airflow dynamics, the hydrostatic limit was considered. This
model version involves one additional nondimensional parameter $\hat{H}$ (Smith and Barstad, 2004, Sect. 3b), where we use $\hat{H} = 1$ suggested as a typical value there. Since the simple upslope version corresponds to $\hat{H} = 0$, the effect of values $\hat{H} \neq 1$ could be estimated qualitatively from the curves. The most important effect of the more elaborate airflow dynamics is that the source term no longer depends only on the local slope at the considered location, but also on the slopes in a larger part of the domain. As a consequence, the precipitation at a given point does not only depend on the topography in the upwind range, but to some
extent also on the topography of the downwind region. This becomes obvious at the plain in front of the ramp, where the precipitation rate already increases at a considerable distance to the ramp. Next, the peak in precipitation is shifted upwind,

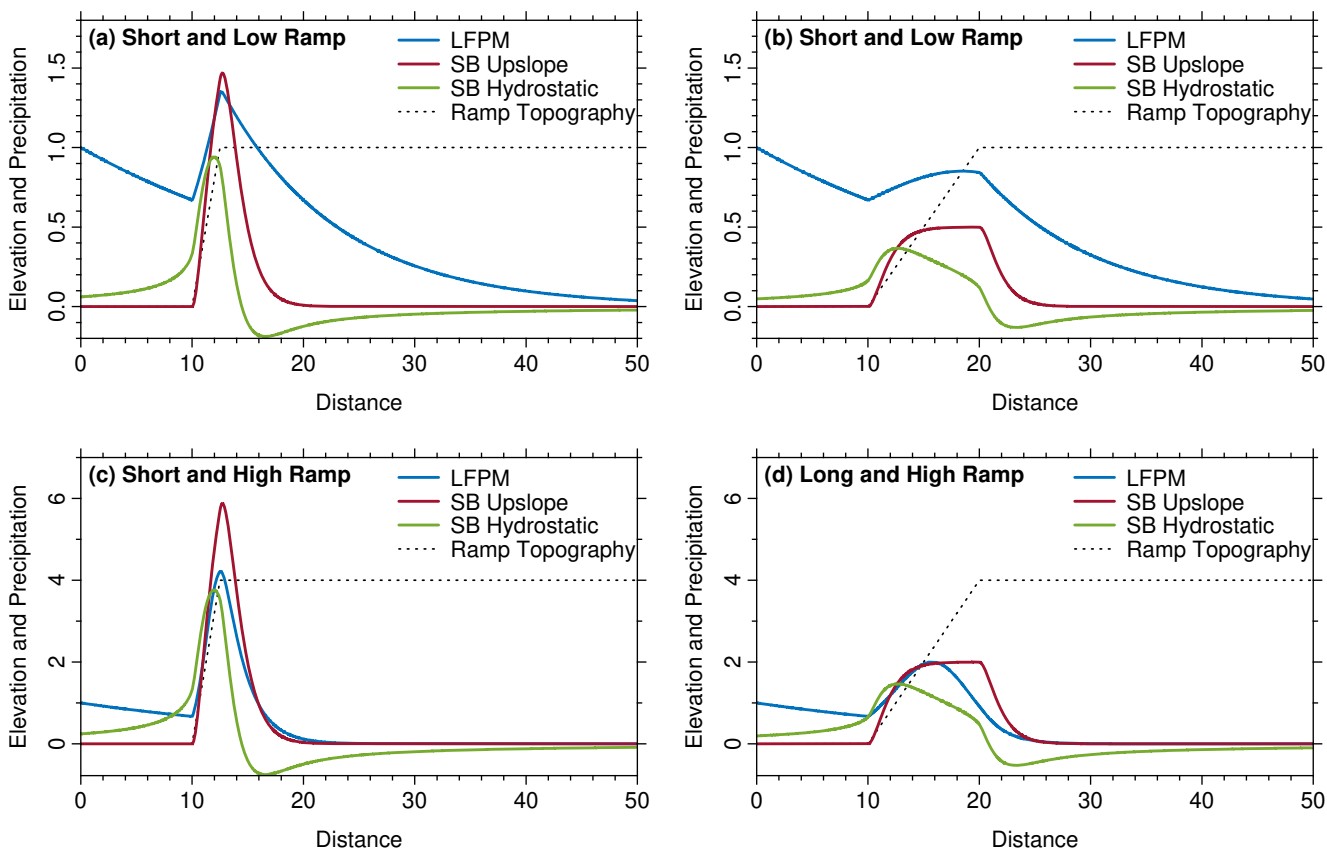

**Figure 5.** Comparison of the LFPM to the simple upslope version of the SB model and the version with advanced airflow dynamics in the hydrostatic limit.

so towards lower elevations at the ramp. Finally, the values become negative at the plateau, which means that the precipitation rate will be lower than the background rate.

Overall, the more elaborate airflow dynamics increases the spatial range over which the topography affects precipitation compared to the simple upslope version of the SB model. However, the precipitation approaches the same value (the background rate) in both directions far away from the ramp for all versions of the SB model, independent of the elevation of the plateau. So none of the versions of the SB model can predict precipitation rates at a high plateau that are much lower than those in the plain in front of the plateau, as it occurs, e.g., in the Tibetan Plateau.

The results obtained from the LFPM are readily understood from the considerations made in the previous sections, where the depletion of moisture by precipitation is the most important phenomenon. If the plateau is low (Fig. 5a,b), the increase in precipitation is moderate. Since the depletion is also moderate then, the plateau is still exposed to considerable precipitation. In turn, precipitation decreases rapidly along the high plateau. The amount of moisture that enters the plateau depends on the

length of the ramp, while the further depletion along the plateau is related to the reduced value of $\beta$ at large elevations. For the long and high ramp (Fig. 5d), the depletion at the ramp is so strong that the entire plateau is quite dry.

Restricted to the region close to the ramp, however, the LFPM and the SB model can be adjusted to yield similar results, although for different reasons. In the LFPM, it is the combination of the elevation-dependent conversion process and the limited amount of moisture, while much of the behavior depends on the model used for the source term in the SB model. The linearity of the SB model also deserves attention in this context. While both models are linear with regard to the water contents, the SB model is also linear concerning the topography. As a consequence, precipitation patterns for the low and high ramps in

Fig. 5 (a/c and b/d) are the same, where only the absolute values scale like the elevation of the plateau. Owing to the linearity, precipitation patterns depend only on the lateral structure of the topography, but not on the absolute elevation in all versions of the SB model. While this can be fixed to some extent by adjusting the model parameters for individual scenarios, it becomes a serious limitation in the context of co-evolution of topography and climate, e.g., if the same uplift pattern is considered at different absolute rates. As it will be shown in Sect. 8, the LFPM is more powerful here.

The linearity of the SB model also affects the precipitation at the leeward side of mountains. The precipitation of a declining ramp would just be opposite to that of an increasing ramp. So if we added a declining ramp behind a large plateau in Fig. 5, the precipitation rates would be negative there. Taking into account that a background rate has to be added to the precipitation rates shown in Fig. 5, this might not be a crucial problem in some of the scenarios, but in particular for short and high ramps (Fig. 5c), the negative rates would be too high to be compensated by the background rate. In order to overcome this problem,

Smith and Barstad (2004) suggested to truncate the precipitation term explicitly at negative values. However, the SB model still predicts extremely dry leeward sides, and improving this behavior was obviously one of the motivations for extending the model by multiple layers proposed by Barstad and Schüller (2011). In contrast, the LFPM does not tend towards extremely dry leeward regions, and it is guaranteed that neither $F_v$ nor $F_c$ can become negative. So the LFPM requires no artificial measures at the leeward side, in particular no truncation in order to avoid negative precipitation rates.

In this sense, including the feedback by re-evaporation in the LFPM may look more complicated first, but it is the key to capturing large-scale precipitation patterns and avoids the need for taking additional measures at the leeward side of mountains.

The model of Garcia-Castellanos (2007) uses a single moisture component and thus seems to be simpler than our approach. The fundamental structure of this model can be explained by considering the limit $L_c \to 0$. This means that the contents of vapor and cloud water immediately achieve an equilibrium $\frac{F_v}{F_c} = \beta$, defined by the elevation-dependent value $\beta$. Adding

Eqs. (10) and (11) then yields

$$-\frac{\partial F}{\partial x} + L_d \frac{\partial^2 F}{\partial y^2} - \frac{F}{(1+\beta)L_f} = 0, \tag{50}$$

where $F = F_v + F_c$ is the total flux per unit width. This is the fundamental structure of the model of Garcia-Castellanos (2007) except for the dispersion term. The precipitation rate is the ratio of $F$ and an elevation-dependent value $(1+\beta)L_f$. The length scale of transport is $L_l = (1+\beta)L_f$ then, which is much larger than $L_f$ if $\beta$ is sufficiently large. So replacing the coupling

between the two flux components by an equilibrium preserves the ability to capture large-scale precipitation dynamics.

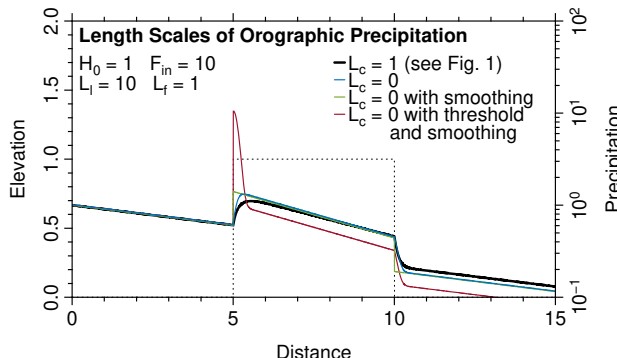

**Figure 6.** Comparison of the LFPM to the one-component model of Garcia-Castellanos (2007). The parameter values are the same as in Fig. 1.

However, the short length scale $L_s$ is lost if we assume instantaneous equilibrium. Precipitation reacts immediately to changes in topography then, so that the precipitation pattern becomes sensitive to the small-scale roughness. In order to overcome this problem, Garcia-Castellanos (2007) introduced an additional smoothing by applying a convolution with half of a Gaussian curve in the upwind direction. As illustrated in Fig. 6, the behavior of the one-component version with additional
485 smoothing is indeed similar to our two-component approach in 1D. Here, a smoothing length $\Delta x = 2L_s$ appears to be a reasonable choice at $H = 0$. However, it should be emphasized that the concept of smoothing including the choice of a smoothing length is an ad-hoc assumption, while smoothing automatically emerges in the LFPM. Beyond this, applying the convolution with half of a Gaussian function is numerically more expensive than the treatment of two moisture components.

In this sense, considering two moisture components is indeed the simplest choice if we want to combine long-range transport
and a smooth response to sharp topographic gradients in a linear model. This result is directly related to the occurrence of two eigenvalues, i.e., horizontal length scales discussed in Sect. 4.1. A single-component model seems to be simpler at first sight, but we have to pay for this simplicity later as soon as we need the second length scale.

Moreover, the model of Garcia-Castellanos (2007) contains a nonlinear component, which is not included in the LFPM. While the precipitation term $\frac{F}{(1+\beta)L_f}$ in Eq. (50) was obtained by simplifying out two-component model, Garcia-Castellanos
(2007) introduced an expression with this structure directly in the form

$$P = P_0 \frac{Q}{Q_{\max}(H)}, \tag{51}$$

where $P_0$ is some reference precipitation, $Q_{\max}(H)$ is an elevation-dependent maximum water content. While this expression is still equivalent to the precipitation term in Eq. (50), it was extended in such a way that Eq. (51) is only applied for $Q < Q_{\max}(H)$. Otherwise, it was assumed that the excess water content $Q - Q_{\max}(H)$ is immediately converted into precipitation.
Transferred to the formalism of Eq. (50), this occurs if the ratio $\frac{F}{1+\beta}$ exceeds a given threshold.

The red curve in Fig. 6 illustrates this effect, where it was assumed that the air at the left-hand boundary is just at the limit $Q = Q_{\max}(0)$. While $Q < Q_{\max}(0)$ in the foreland, $Q_{\max}(H)$ (via $1 + \beta$ in our formalism) decreases suddenly at $x = 5$ due

to the sudden increase in $H$, so that $Q > Q_{\max}(H)$. This would even result in a sharp peak in the precipitation curve without smoothing. Smoothing in the upstream direction reduces the height of the peak and widens it into the plateau.

As a main effect of this extension, the model of Garcia-Castellanos (2007) becomes able to predict some kind of overshooting in precipitation at the windward side of mountains. For a high plateau, the original version predicts a gentle increase in precipitation to the level at the plateau. A decrease in precipitation only occurs due to the decrease in water content or, if we consider a mountain range, due to decreasing elevation at the leeward side.

     Such an effect might be useful, although the physical basis is not trivial since a dynamic equilibrium between vapor and cloud
water is already included in the linear model. The LFPM could be extended by nonlinearity in several ways. The precipitation process may be a candidate here since coagulation plays a part in the growth of hydrometeors, and the rate of coagulation increases rather quadratically than linearly with concentration. However, following the concept of parsimony, we do not follow ideas of nonlinearity further in this study.

## 7    A real-world example

This section presents an application of the LFPM to a real topography. It should, however, rather be seen as an illustration than as a validation. While the attempt to validate the SB model by Barstad and Smith (2005) suffered from the availability of data at a sufficient spatial resolution, we must keep in mind that all models discussed in this paper were not tailored for reproducing precipitation patterns exactly. As discussed in Sect. 6, predicting the effects of changes in topography on precipitation is more important and more challenging in the context of landform evolution than, e.g., adjusting a model to predict the precipitation
rate at the front of a mountain range close to the coast. Thus, a serious validation would have to be based on several locations and scenarios, where the LFPM and other models would have to be tested against real-world data and regional climate models.

     Figure 7 compares the precipitation pattern of the India-Asia collision zone modeled with the LFPM to the annual precipitation pattern of the TRMM2b31 data set (Bookhagen and Burbank, 2010) and the WorldClim2.1 precipitation data (Fick and Hijmans, 2017). The TRMM2b31 dataset is an outcome of the Tropical Rainfall Measurement Mission, where the average
precipitation rates are based on a 12 years (1998 to 2009) time series. The dataset has a spatial resolution of roughly 5 km and covers the region between $36°$S and $36°$N. The WorldClim2.1 dataset is a global dataset of climate variables with spatial resolutions between 10 arc minutes ($\approx 18$ km) and 30 arc seconds ($\approx 0.9$ km). Monthly precipitation data are averaged for the years from 1970 to 2000 and based on a large number of weather stations. Integrating the average precipitation from January to December results in the annual precipitation dataset shown in Fig. 7d.

Simulations of the LFPM were performed on a regular grid with 1 km mesh width for two different directions of atmospheric flow (from south to north and from south-west to north-east). Ocean areas were considered as boundaries, where a uniform influx was assumed. The domain was extended in such a way that each flow line (either from south to north or from south-west to north-east) starts from a point in the ocean.

     Several simulations with different parameter values were conducted. However, it is immediately recognized in Fig. 7 that
the flow direction has a strong influence on the precipitation pattern. Compared to this influence, the effect of the parameter

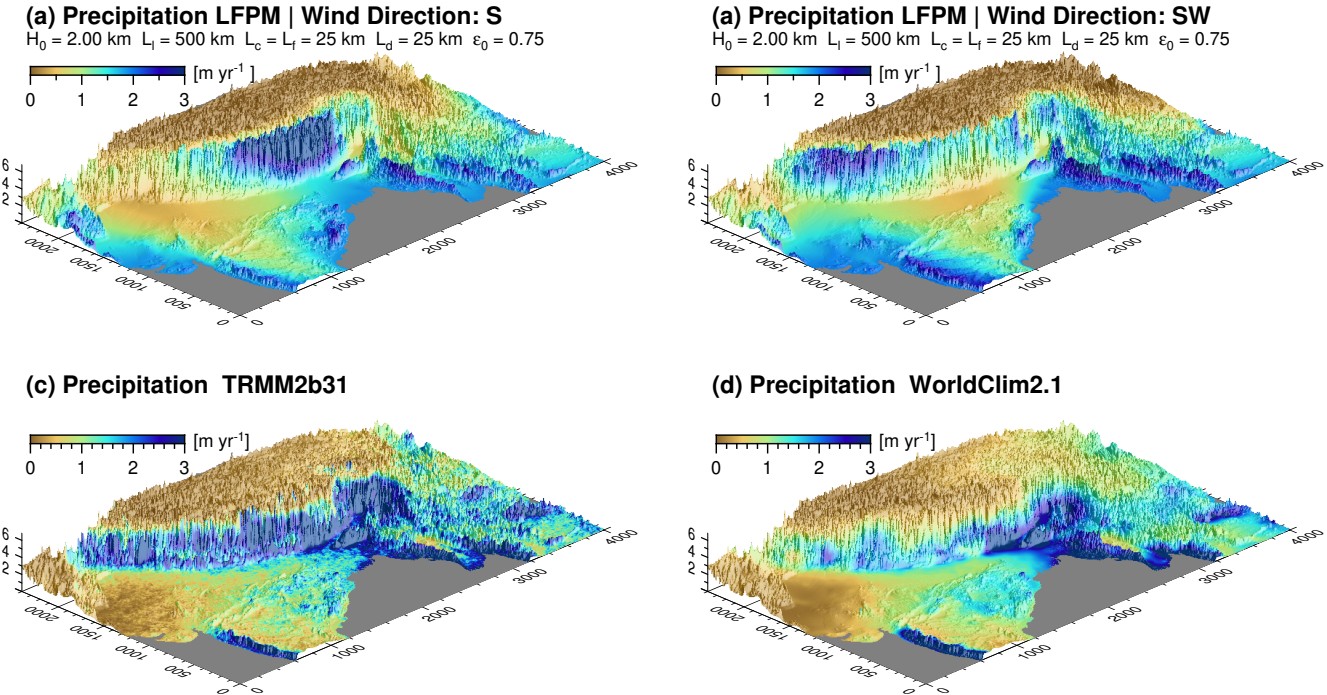

**Figure 7.** Comparison of precipitation patterns of the India-Asia collision zone. Precipitation modeled with the LFPM assuming wind from (a) south and (b) southwest. (c) TRMM-based precipitation data (Bookhagen and Burbank, 2010), and (d) precipitation data from the WorldClim2.1 dataset (Fick and Hijmans, 2017).

values is much weaker. In particular, similar results in terms of the root mean square (RMS) deviation were found for different combinations of the parameter values. So a formal fit would not be very useful. Instead, $L_c = L_f = 25$ km and $L_l = 500$ km were chosen as reasonable values according to the considerations in the previous sections. A rather high evaporation ratio $\epsilon_0 = 0.75$ was assumed, while lower values would yield similar results with greater values of $L_l$ according to the findings of

Sect. 5. Finally, $H_0 = 2$ km and $L_d = 25$ km were assumed. The influx at the boundaries was adjusted automatically in such a way that the average precipitation over the domain matches the respective average TRMM2b31 precipitation (0.99 m yr$^{-1}$), which is very close to the average WorldClim2.1 precipitation (1.00 m yr$^{-1}$).

While some large-scale properties of the precipitation pattern such as wet regions at the western coast of India (Ghats escarpment) and at the orographic front of the Himalaya and the dry Tibetan Plateau are at least qualitatively reproduced by the LFPM,

deviations in precipitation rate between the different datasets are distinct. The root-mean square (RMS) deviation towards the TRMM2b31 dataset for the given parameters with a wind direction from south and from southwest amount to 0.81 m yr$^{-1}$ and 0.84 m yr$^{-1}$, respectively. The RMS deviations towards the Worldclim2.1 dataset are slightly lower (0.77 m yr$^{-1}$ and 0.80 m yr$^{-1}$).

The RMS deviations can be reduced to about 0.6 m yr$^{-1}$ by tuning the parameters. In particular, increasing $H_0$ goes along
with higher precipitation rates in the Tibetan Plateau (closer to the measured data) and lower precipitation rates at the windward
side of topographic barriers. While this tuning reduces the RMS deviation, it does not necessarily result in a better reproduction
of the most striking precipitation features such as high orographic precipitation at the windward side and rain shadows at the
leeward side. Furthermore, large values of $H_0$ would lead to very small orographic effects in other, much lower mountain
ranges.

In general, we should be careful not to compensate principal limitations of the model by potentially unrealistic parameter
values. In particular, this applies to the pre-defined uniform wind direction in the LFPM, which cannot describe the atmospheric
circulation pattern of the entire region sufficiently well. Furthermore, obvious differences occur also between the TRMM2b31
and the Worldclim2.1 dataset. Compared with the WorldClim2.1 data, the TRMM2b31 data indicate much higher precipitation
rates at the western Himalaya, but distinctly dryer conditions at the eastern Tibetan Plateau. These differences suggest that
the precipitation rates still involve a considerable uncertainty. The RMS deviation between these two datasets is 0.42 m yr$^{-1}$,
which is not far below the deviation of our "best fit" parameter set.

## 8   Examples of co-evolution of topography and climate

Similarly to the approaches of Roe et al. (2003), Smith and Barstad (2004), and Garcia-Castellanos (2007), the scope of the
model developed in this study is not a precise prediction of precipitation rates, but its combination with long-term landform
evolution. This section provides some examples exploring the effect of orographic precipitation and continentality on fluvial
landform evolution and the feedback of the resulting topography on the precipitation pattern.

As described in Sect. 1, the SPIM and its derivates can easily be extended by a variable effective precipitation and thus be
coupled with the LFPM. In the following, we use a model that is not restricted to pure bedrock incision, but also takes into
account sediment transport. While the idea behind this model dates back to Whipple and Tucker (2002) and Davy and Lague
(2009) or partly even to older studies (Howard, 1994; Kooi and Beaumont, 1994), it is used here in the most recent formulation,
the so-called shared stream-power model (Hergarten, 2020). The constitutive equation of the shared stream-power model reads

$$\frac{E}{K_\mathrm{d}} + \frac{Q}{K_\mathrm{t} A} = A^m S^n, \tag{52}$$

where $Q$ is the sediment flux (volume per time; not to be confused with the atmospheric water contents $Q_\mathrm{v}$ and $Q_\mathrm{c}$). This model
contains two erodibilities, where $K_\mathrm{d}$ describes the erodibility in absence of transported sediment, while $K_\mathrm{t}$ characterizes the
ability to transport sediment at zero erosion. For a deeper insight into the properties of the shared stream-power model and the
meaning of its parameters, the reader is referred to Hergarten (2021b).

For spatially uniform erosion, the sediment flux is $Q = EA$, and Eq. (52) collapses to a form analogous to the SPIM (Eq. 1)
with an effective erodibility $K$ according to

$$\frac{1}{K} = \frac{1}{K_\mathrm{d}} + \frac{1}{K_\mathrm{t}}. \tag{53}$$

Following Robl et al. (2017), we use $n = 1$ (i.e., the linear version of the model), $m = 0.5$, and $K = 2.5$ Myr$^{-1}$. Studies of natural and experimental river profiles at the transition to a foreland by Guerit et al. (2019) suggest a ratio $\frac{K_d}{K_t} \approx 1.6$ ($G$ in their notation) for $n = 1$, which leads to $K_d \approx 6.5$ Myr$^{-1}$ and $K_t \approx 4.1$ Myr$^{-1}$.

The concept of expressing discharges as their catchment-size equivalent (referring to a hypothetic uniform reference precipitation rate) is particularly useful in the context of the shared stream-power model. If both occurrences of $A$ in Eq. (52) are interpreted as catchment-size equivalents of the actual discharge, Eq. (52) remains formally the same, including the values and the units of $K_d$ and $K_t$.

We use a regular mesh of $2000 \times 2000$ nodes for a domain of 500 km linear size, corresponding to a spatial resolution of 250 m. Although rather coarse, hillslope processes are still relevant at this scale. Using a purely fluvial model would lead to artificially steep slopes and thus increased elevations at the drainage divides, which may affect the precipitation pattern despite the robustness of the model against the small-scale roughness of the topography. In order to avoid this, we use an approach that was brought into play in the context of debris flows in steep valleys by Stock and Dietrich (2003) and developed further by Hergarten et al. (2016). This approach replaces the term $A^\theta$ in Eq. (1) (or here in Eq. 52) by $A^\theta + A_c^\theta$, where $\theta = \frac{m}{n}$ and $A_c$ is a given constant. This modification acts like an increased catchment size or like an increased discharge and thus avoids the occurrence of extremely steep slopes at small catchment sizes. We use $A_c = 1$ pixel $\approx 0.06$ km$^2$ here, which roughly consistent with the estimate $A_c = 0.05$ km$^2$ obtained by Hergarten et al. (2016) for the topography of Taiwan.

In the following section, we show how the decrease in precipitation rate with distance to the source of moisture (i.e., continentality) controls the shape and the height of large mountain ranges. Then we illustrate topographic patterns resulting from feedbacks between rock uplift, orographic precipitation, and fluvial erosion.

## 8.1 Impact of continentality on landform evolution

The large-scale precipitation pattern over continental areas is controlled by the length scale of long-range moisture transport $L_l$. If the extension of a mountain belt in wind direction reaches the order of magnitude of $L_l$, the precipitation pattern may have a strong influence on its height and shape even without any immediate effect of elevation on precipitation. In terms of the model parameters, this situation is described by a reference elevation $H_0$ much larger than the surface elevation. Then the precipitation pattern reflects increasing continentality with an exponential drop in precipitation rate from the moisture source towards the continental inland. The precipitation rate is solely controlled by the influx $F_{in}$ and the length scale $L_l$, which is further stretched by considering evaporation.

Figure 8 shows the effect of continentality on topography. The considered mountain range is 300 km wide and uplifted at a rate of 0.25 km Myr$^{-1}$ in all three examples. The two 100 km wide foreland regions are tectonically inactive. Moisture enters the model domain at the southern boundary and is advected towards north. While the geometry, the length scales $L_c = L_f = 25$ km, the elevation-independent evaporation ratio $\epsilon = 0.5$, and the amount of moisture entering the southern boundary are the same in all three scenarios, the length scale $L_l$ of the long-range transport varies from 50 km to 600 km. Only steady-state topographies are considered, where the term steady state is used in a sloppy way here since the drainage pattern in an inactive

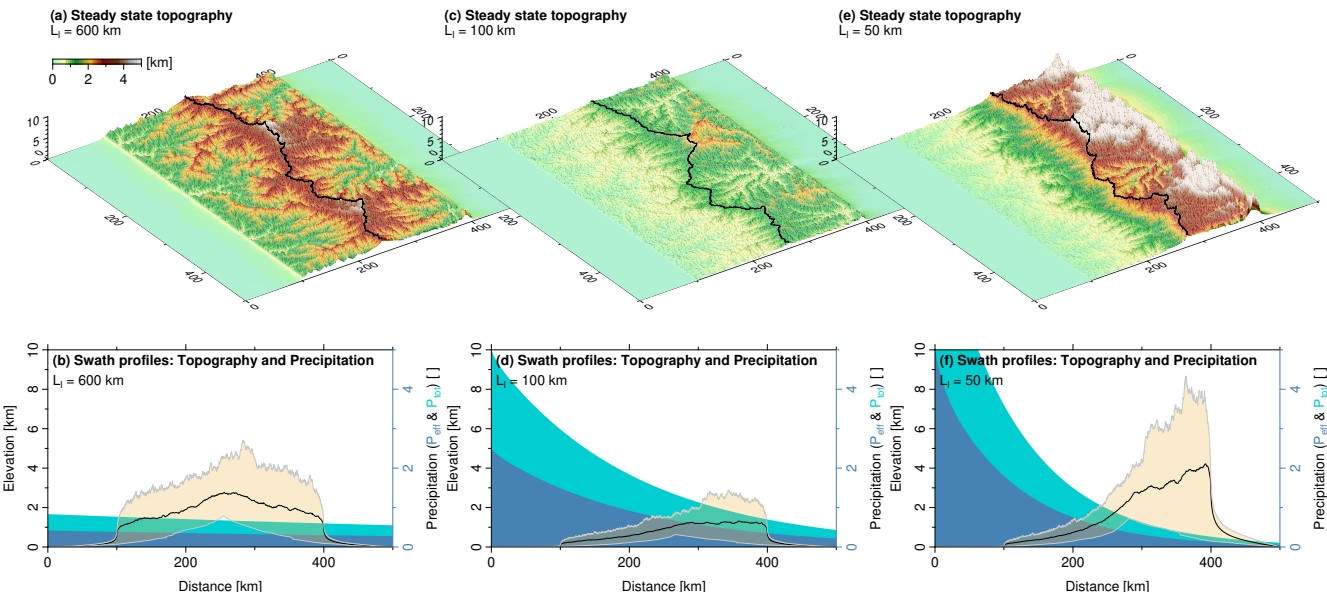

**Figure 8.** Continentality and steady-state mountain range geometry. Uplift rate ($U = 0.25$ km Myr$^{-1}$), erodibility ($K = 2.5$ Myr$^{-1}$) and factor of evaporation ($\epsilon = 0.5$) are the same in all three experiments. (a, c, e) show perspective views of steady state mountain topography for $L_l = 50$ km, 100 km and 600 km, respectively. The black solid line indicates the main drainage divide separating the windward from the leeward side of the mountain range. (b, d, f) are south-north trending swath profiles representing total precipitation (turquoise area) and effective precipitation (blue area) and topography (brown) with average (thick solid line) and extreme values (thin solid lines) taken over the full west-east extent of the model domain.

foreland permanently reorganizes (Yuan et al., 2019). Since this reorganization has a minor effect on the mountain topography,

the mountain range comes close to a steady state with some fluctuations.

Two effects of continentality must be distinguished. First, $L_l$ has an influence on the amount of precipitation that the mountain range receives in total. This amount should affect the total height of the steady-state mountain range. It is small if $L_l$ is small since most of the moisture is already lost in the foreland, but also small if $L_l$ is large since most of the moisture passes the domain without much precipitation then. So there must be a length $L_l$ where the amount of precipitation on the mountain range

becomes maximal.

For $L_l = 600$ km (Fig. 8a,b), evaporation with $\epsilon = 0.5$ effectively stretches $L_l$ to about 1200 km following the considerations of Sect. 5. Since this is four times the width of the mountain range, the precipitation rate varies by a factor of $e^{\frac{1}{4}} \approx 1.28$ over the mountain range. Differences in discharge are, however, smaller. If the drainage divide is in the middle, the difference in total precipitation differs only by a factor of $e^{\frac{1}{8}} \approx 1.13$ between the windward and the leeward half of the mountain range. So

the discharges of the big rivers only differ by this factor at the edges of the mountain range.

This difference is visible in the swath profile (Fig. 8b). The distribution of maximum elevations across the mountain range is already quite asymmetric since small catchment sizes at ridge lines and hillslopes cause the erosion rate to be directly related to

the local precipitation rate. Hence, the highest domains become increasingly steeper towards the leeward side. In contrast, the minimum elevation of the swath profiles describes the large rivers, which differ not so much in their discharge. So the profile of the minimum elevation is still quite symmetric here. This also implies that the relative incision of the rivers in relation to the hillslopes is deeper at the leeward side than at the windward side.

According to the findings of Sect. 5, the length scale $L_1 = 100$ km considered in Fig. 8(c,d) is stretched by evaporation to 206 km. This value is close to the length scale where the mountain range receives the maximum amount of precipitation in total, $\frac{300 \text{km}}{\ln 4} = 216$ km. Consequently, the overall height of the mountain range is quite low here. Since precipitation varies by a factor of $e^{\frac{300}{206}} \approx 4.3$ here, the topography becomes strongly asymmetric. Again, this asymmetry mainly concerns the maximum elevation in the swath profile (Fig. 8d), while the asymmetry of the minimum elevation referring to the largest rivers is smaller. Nevertheless, the asymmetry in the minimum elevation is clearly visible here, and it goes along with a shift of the principal drainage divide towards the leeward side. While the total area drained by the leeward part of the mountain range was 48 % of the total area of the mountain range for $L_1 = 600$ km, this fraction has decreased to 42 % now. Despite this moderate shift, the highest peaks are already separated from the main drainage divide. For $L_1 = 50$ km (Fig. 8e,f, effectively 114 km with evaporation), the total amount of precipitation on the orogen decreases. This results in an increasing overall surface elevation. More importantly, the topography becomes extremely asymmetric since the precipitation rate varies by a factor of $e^{\frac{300}{114}} \approx 14$ over the mountain range. This variation is strong enough to make the river profiles (minimum elevation in Fig. 8f) strongly asymmetric. The overall asymmetry is so strong that it also dominates the mean elevation. Apart from very high peaks close to the leeward border of the mountain range, the highest mean elevation is also achieved there, while the version with $L_1 = 100$ km featured an almost constant mean elevation over the leeward part of the mountain range. In turn, the shift of the main drainage divide is only moderate compared to the previous scenario. The leeward fraction of the total drained area has decreased from 42 % to 39 %. As a consequence, the large massifs that have formed in the northern part drain almost entirely towards the leeward side. So rivers starting from high regions partly drain towards south first, but then change their flow direction towards the large north-trending valleys.

## 8.2 Orographic precipitation controlling mountain range geometry

We finally consider the effect of topography on the precipitation pattern and the resulting feedback on landform evolution. The overall geometry and the parameter values are the same as before, except for a fixed length scale $L_1 = 500$ km, which was chosen in such a way that the effect of continentality over the mountain range is rather weak. In contrast to the previous examples, transversal dispersion is relevant here, where $L_d = 5$ km was chosen.

While the vertical length scale could be defined arbitrarily based on the erodibility and the uplift rate in the previous examples, it is defined here by the reference elevation $H_0$ that describes the decrease of $\beta$ and $\epsilon$ with elevation. We use a fixed value $H_0 = 1$ km and consider scenarios with different uplift rates.

The results shown in Fig. 9 reveal a distinct difference not only in mountain height, but also in mountain range asymmetry and spatial gradients in precipitation rate. As expected, mean and peak elevation increase with uplift rate. In strong contrast to scenarios of uniform precipitation, the highest mean and peak elevations are shifted towards the leeward side of the moun-

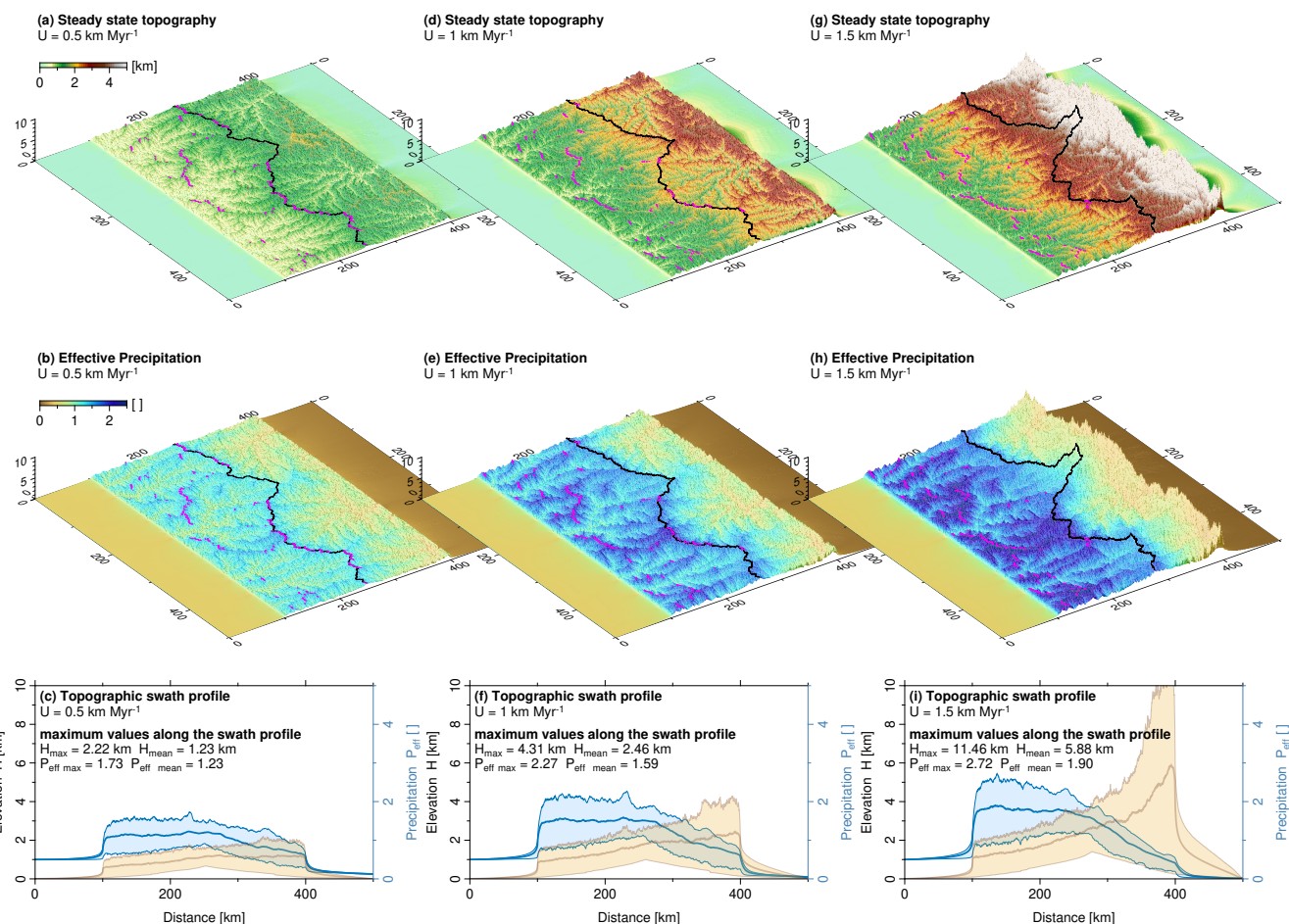

**Figure 9.** Impact of uplift rate on the precipitation pattern and on mountain geometry for $K = 2.5$ Myr$^{-1}$ and $L_l = 500$ km, $L_c = L_f = 25$ km, $L_d = 5$ km and $\epsilon = 0.5$. (a, d, g) show steady-state mountain topographies representing uplift rates of 0.5 km Myr$^{-1}$, 1.0 km Myr$^{-1}$ and 1.5 km Myr$^{-1}$. The black solid line indicates the main drainage divide. Magenta dots mark the position of maximum precipitation in windward direction. (b, e, h) show the corresponding precipitation patterns and (c, f, i) are south-north trending swath profiles representing topography (brown) and effective precipitation (blue) with average (thick solid line) and extreme values (thin solid lines) taken over the full west-east extent of the model domain.

tain range. The observed asymmetry with a gentle increase in elevation on the windward side and a strong decrease on the leeward side increases with uplift rate. In all scenarios, the highest rates of effective precipitation occur at the windward side of the mountain range, but these spatial gradients in precipitation rate increase distinctly with uplift rate. At an uplift rate of 665   1.5 km Myr$^{-1}$, the average effective precipitation decreases by a factor of about five from the windward to the leeward mountain front (Fig. 9i). The principal drainage divide is shifted towards the leeward side with increasing precipitation gradient and uplift rates, but to a much lesser extent than the distribution of mountain heights would suggest.

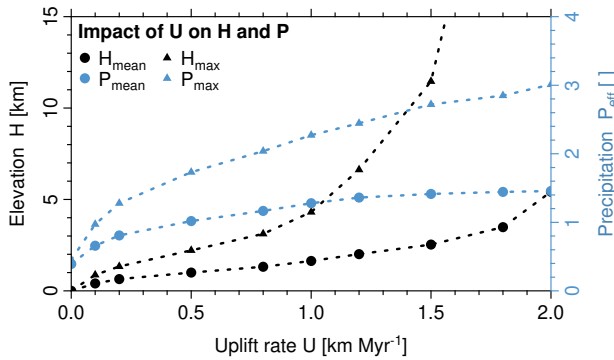

**Figure 10.** Uplift rate control on precipitation rate and mountain height. Mean and maximum values for precipitation and elevation are computed for the domain of the mountain range only.

As illustrated in Fig. 10, the relationship between uplift rate and mountain height becomes nonlinear in contrast to simple scenarios assuming a uniform precipitation rate. At small uplift rates, the behavior is dominated by the increase in precipitation with increasing topography, which results in higher erosion rates. As a consequence, the topography still increases with uplift rate, but the increase is weaker than linear. For the scenario considered here, this holds for uplift rates of up to about $0.5$ km Myr$^{-1}$.

The concavity of the relation between uplift rate and topography is lost at higher uplift rates. The limited amount of moisture supplied from the boundary plays a central part here. As a consequence, the mean effective precipitation rate approaches a constant value for large uplift rates. However, even a constant mean precipitation rate does not imply a linear relation between uplift rate and topography since the spatial distribution of the precipitation becomes increasingly inhomogeneous. This effect is recognized in the maximum precipitation rate in Fig. 10, which continuously increases with increasing uplift.

Figure 11 illustrates the relation between topography and precipitation for the three considered uplift rates. A bimodal distribution is found at low elevations. As shown in Fig. 9(c,f,i), low elevations occur along big valleys close to the boundaries of the mountain range. Since precipitation decreases systematically from the windward side to the leeward side, the low-elevation range splits up into a rather wet windward domain and a rather dry leeward domain. This distinction is, however, lost with increasing surface elevation since intermediate elevations are distributed over the entire mountain range. This goes along with a rapidly increasing variability in precipitation at given elevation.

While some decline of the increase in precipitation with elevation is already visible for $U = 0.5$ km Myr$^{-1}$, it even turns into an absolute decrease at large elevation. The highest precipitation rates are found at $H \approx 2$ km for $U = 1$ km Myr$^{-1}$ as well as for $U = 1.5$ km Myr$^{-1}$ and decrease above this elevation. This decrease is not an immediate effect of the elevation since the model itself predicts a continuous increase in precipitation with elevation at a given moisture content. As discussed above, it arises from the limited amount of moisture supplied from the windward boundary. So the occurrence of the highest precipitation rates at $H \approx 2$ km is not only related to the parameters of the precipitation model, but also to the properties of the erosion model. As a further consequence, an extreme variation in precipitation occurs at $H \approx 2$ km with high rates at the

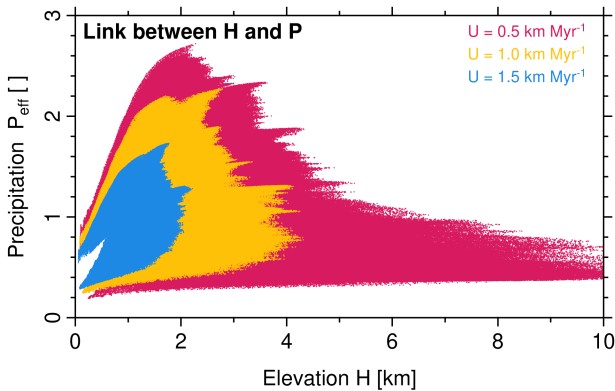

**Figure 11.** Relationship between surface elevation and precipitation for varying uplift rates (scenarios are shown at Fig. 9).

windward side and very dry regions in the shadow or high mountains where most of the available moisture has already been consumed.

The decrease in precipitation at large elevations results in a strong interaction with landform evolution. Since the erosion rate at a given channel slope decreases then, an equilibrium between uplift and erosion can only be achieved by increasing the channel slopes. Since this also requires increasing elevations, a positive feedback occurs. This feedback is visible as a convex relation between uplift rate and elevation in Fig. 10, which means that elevation increases stronger than linearly with uplift rate. This effect is particularly strong in the maximum elevation. It corresponds to the formation of very high peaks close to the leeward boundary of the mountain range (Fig. 9e,i), while the major valleys between these peaks are not particularly high.

While the precipitation pattern explains several properties of the resulting topography, we should keep in mind that the erosion rate depends on the discharge and not on the local precipitation rate. A consequence of this difference is recognized in the leeward foreland in Fig. 9. While the precipitation rate is overall low here, the topography becomes highly variable for $U = 1.5 \, \text{km} \, \text{Myr}^{-1}$.

Huge alluvial fans form behind the highest regions (at $x \approx 200$ km and at $x \approx 400$ km). Their occurrence is related to the low precipitation in the northern foreland. Figure 12(a) reveals that the respective catchments are quite small and completely in the precipitation shadow of the large massifs. This results in a very low discharge and thus in a limited ability to transport the sediment coming from the high mountain region. In equilibrium, the low discharge must be compensated by a high channel slope, which leads to the formation of the huge alluvial fans.

The regions between these fans feature rivers with large catchments, which reach deep into the mountain range up to the principal drainage divide. Since a part of the moisture coming from the windward boundary passes the principal drainage divide, parts of these catchments are exposed to high precipitation. As a consequence, the discharges are rather high, although the precipitation rate in the leeward foreland is overall low. Therefore, the respective rivers are able to carry the sediment coming from the mountain region without forming large alluvial fans.

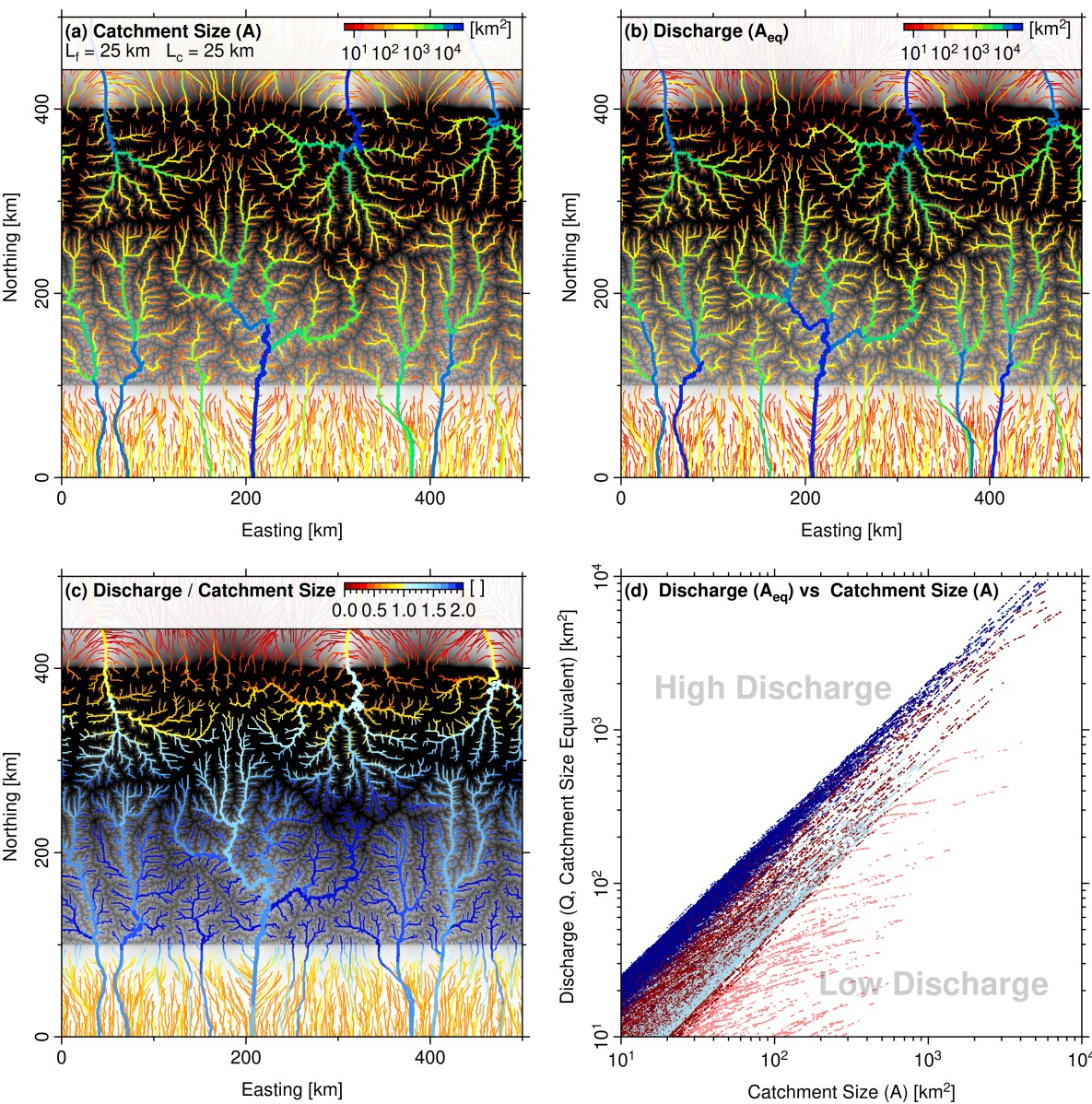

**Figure 12.** Spatial distribution of precipitation and discharge for $U = 1.5$ km Myr$^{-1}$, $K = 2.5$ Myr$^{-1}$, $L_l = 500$ km, $L_c = L_f = 25$ km, $L_d$ = 5 km, and $\epsilon = 0.5$. Topography and precipitation pattern of this scenario is shown on Fig. 9 (right-hand column). Channels are color-coded for (a) contributing drainage area and (b) river discharge as catchment-size equivalent $A_{eq}$. (c) ratio of discharge $A_{eq}$ and catchment size $A$. All channels with a catchment size $A \geq 25$ km$^2$ are considered. (d) Relationship between catchment size and discharge at the windward (blue dots) and the leeward (red dots) side of the mountain range. Dark colors represent the mountain range and light colors the forelands.

Figure 12(c) shows that the windward foreland also features rivers with strongly different ratios of discharge and catchment size. While the reason for this variation is basically the same as in the leeward foreland, the variation is less pronounced here since the windward foreland region is exposed to a higher precipitation than the leeward foreland. Concerning the resulting topography, however, the main difference between the two foreland regions consists in the presence of low-discharge rivers bringing sediments from the mountain range to the leeward foreland. In the windward foreland, rivers with low discharge to catchment-size ratio typically have their source in the foreland itself and thus carry only a small amount of sediment. Therefore, the windward foreland features no big alluvial fans.

As shown in Fig. 12(d), the difference in the discharge characteristics between the leeward side and the windward side is not restricted to the foreland regions. The discharge at any given catchment size $A \geq 10 \ \mathrm{km}^2$ varies by less than a factor of 2 at the windward side, while the variation at the leeward side is higher.

Following individual rivers downstream, the change in the discharge characteristics is opposite in both domains. It is recognized in Fig. 12(c) that rivers originating close to the principal drainage divide start with similar discharge to catchment-size ratios at both sides. However, tributaries at the windward side are exposed to higher precipitation, so that the discharge to catchment-size typically increases downstream at the windward side until the rivers reach the foreland. In turn, tributaries at the leeward side are typically quite dry and thus cause a downstream decrease in the discharge to catchment-size ratio. As a consequence, equilibrium profiles of tributaries at the leeward side are rather steep compared to the main rivers, which corresponds to a deeper incision of the large river valleys than at the windward side.

## 9  Scope, limitations, and perspectives

After performing some simple comparisons of the LFPM with existing models and exploring its potential in the context of landform evolution modeling, we now recapitulate what the LFPM is and what it is not.

As discussed in Sect. 6, the LFPM has some advantages over the models proposed by Smith and Barstad (2004) and by Garcia-Castellanos (2007). However, all these models share the restriction to pre-defined atmospheric flow patterns. This restriction reduces not only the numerical complexity extremely in comparison to regional climate models, but also the predictive power.

At the actual level, models of this type are particularly useful for theoretical considerations of the co-evolution of topography and climate. There are numerous open questions not only about the overall asymmetry of mountain ranges, but also about asymmetries at smaller scales (particularly individual drainage divides, e.g., Trost et al., 2020) or the longevity of large plateaus. In a first step, such theoretical studies, where typically artificial topographies are used under well-defined conditions, aim at a principal understanding. The second step, however, is finding out what topographic signatures can tell about the climatic conditions in the past.

When leaving the realm of theoretical studies and sensitivity experiments and looking at real orogens in the geologic past, additional challenges arise. Large changes in topography do not only affect the precipitation pattern within an orogen, but may even change the global climate. As an example, Takahashi and Battisti (2007) found that the Andes play a central part for the

climatic north-south asymmetry around the equator. So at least the input of moisture at the windward boundary and probably also the pre-defined direction of the atmospheric fluxes would have to be taken into account, e.g., by coupling the model to a general circulation model (GCM).

While general circulation models have played a central part in palaeoclimate reconstructions, the recent study of Mutz and
750 Ehlers (2019) focuses on properties that are particularly relevant for Earth surface processes. This direction of research can be seen as a first attempt to approach the co-evolution of topography and climate from large scales. So it is somehow complementary to theoretical studies and sensitivity experiments with landform evolution models. Coupling GCMs with landform evolution models and simple models of orographic precipitation such as the LFPM might become a point where the two complementary approaches meet in the future.

## 10 Conclusions

This study presents a new model for orographic precipitation for use in large-scale landform evolution models such as the SPIM or the shared stream-power model. The goal was to arrive at a model that goes clearly beyond the simplest concepts such as predicting the precipitation rate directly from surface elevation or local slope, but to stay at a level of complexity consistent with simple landform evolution models. In particular, the numerical complexity should not be much higher than that of the
760 respective landform evolution models.

The linear feedback precipitation model (LFPM) developed in this study describes two moisture components, which are interpreted as vapor and cloud water. In contrast to previous models used in this context, a two-way conversion between both components was assumed without considering a thermodynamic equilibrium explicitly. While this concept, where an equilibrium evolves dynamically, seems to be more complicated first, it helps to navigate around some problems and requires
little further assumptions.

As a key property, the LFPM captures a decrease in precipitation with increasing distance from the ocean (or any other source of moisture). This decrease is very slow over large continental areas with little topography, but becomes faster if orographic precipitation at large mountain ranges consumes a considerable part of the available moisture. While precipitation overall increases with elevation in the model, it may decrease again at high elevations due to the limited amount of moisture. As a
770 second important property, precipitation responds to changes in topography not instantaneously, but with a finite length scale. Therefore, the model is not sensitive to the small-scale roughness of the topography and can be operated without any additional smoothing.

The length scale of the decrease in precipitation due to the limited amount of water and the length scale of the response to changes in topography can be computed from the velocity of transport in the atmosphere and the time scales of the conversion
between vapor and cloud water and of the fallout of precipitation. The model structure proposed here is in principle the minimum model that is able to reproduce a long-range transport and a response to changes in topography with a finite length scale.

The model also includes dispersion of the moisture fluxes in direction perpendicular to the main transport direction. This component of the model is particularly useful in combination with two-dimensional landform evolution models since precipitation shadows of infinite length would occur behind individual peaks otherwise. Numerically, the dispersion is the most expensive part of the model. However, it can be implemented as a series of one-dimensional diffusion problems as long as the main direction of transport follows one of the principal coordinate axes. The numerical complexity is still linear then, which means that the computing effort increases only linearly with the total number of nodes of the grid. Since contemporary, fully implicit numerical schemes for the respective erosion models are also of linear complexity, this property is essential for preserving the high numerical efficiency of these models.

The model can easily be extended by a simple model of evapotranspiration where an elevation-dependent fraction of the precipitation is returned to the atmosphere. While this extension increases the length scale of long-range transport further, it does not change the properties of the model fundamentally.

Even the simple examples presented in this study show the remarkable impact of continentality and orographic precipitation on mountain range geometry and on the co-evolution of topography and precipitation pattern. Future studies can use this numerically efficient approach to address a wide range of research questions in the field of landscape evolution where the assumption of uniform precipitation is too simple to explain landscape metrics and topographic patterns.

*Code and data availability.* Codes and results of the simulations are available at the FreiDok data repository 219131 (Hergarten and Robl, 2021). The open-source landform evolution model OpenLEM is freely available at http://hergarten.at/openlem, which also contains an implementation of the LFPM. Interested users are advised to download the most recent version of OpenLEM or the respective standalone version of the LFPM. The authors are happy to assist interested readers in reproducing the results and performing subsequent research.

*Video supplement.* The video supplement includes the respective time-dependent simulations of Sect. 8.2 and a reduced version of Fig. 12 optimized for color-blind readers.

*Author contributions.* S.H. developed the theory and the main parts of the codes. J.R. performed the simulations and analyzed the data. Both authors wrote the paper.

*Competing interests.* The authors declare that they have no conflict of interest.

*Acknowledgements.* The numerical simulations were performed on the Doppler HPC-facility of the University of Salzburg. All figures were generated with the Generic Mapping Tools (Wessel et al., 2019). The authors would like to thank Sebastian Mutz, Travis A. O'Brien, and Kyungrock Paik for their constructive comments.

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
