# Peer review of "The linear feedback precipitation model (LFPM 1.0) – a simple and efficient model for orographic precipitation in the context of landform evolution modeling"

_Geoscientific Model Development, 2021_

## Author Response (AR1)

Dear Reviewers, dear Editor,

thank you very much for your constructive and encouraging comments. The points addressed in the two reports are discussed below, where changes to the manuscript are highlighted in bold letters. Line numbers refer to the version with highlighted changes.

Best regards,

Stefan Hergarten und Jörg Robl

**Reviewer 1 (Kyungrock Paik)**

*Co-evolution of topography and local climate is a hot subject, and a numerical modeling approach as attempted in this paper is highly anticipated. I found this paper interesting and well-written. I see a great potential contribution of this paper to the community. Nevertheless, basic questions remain as follows, for the submitted manuscript.*

*I first am a bit uncertain about the 'main' focus of this study. Is this to propose a new orographic precipitation model? Or do authors put focus on co-evolution? I think this has to be clarified first. I have different comments depending on the choice. If the former is the focus (seemingly from the title of the manuscript), I feel that it requires model comparison and validation with observed data. Earlier orographic rainfall models (e.g., Smith and Barstad, 2004 cited in this work) have done this job thoroughly. By contrast, there is no single comparison with a real precipitation field. This is something missing and requires some work.*

Focus of our paper is – of course – on the new model approach for orographic precipitation. What else should be the reason for spending 50 equations on developing the model and investigating its properties? The potential field of application is indeed modeling the co-evolution of topography and climate since all models of these type are too simple for a "realistic" application to natural topographies.

In this context, we disagree to your statement that the model of Smith & Barstad (2004) was validated thoroughly. The first paper ended at a somehow realistic precipitation pattern, but to be honest, it was not much more than a stronger precipitation at the windward side compared to the leeward side. The subsequent study (Barstad & Smith 2005, doi 10.1175/JHM-404.1) mainly revealed that a validation is difficult (or at least was difficult with the data available at the time). Even in the later paper by Barstad & Schüller (2011, doi 10.1175/JAS-D-10-05016.1), which introduced layers mainly to overcome the problems of the model at the leeward side, the only real-world example was not very convincing at the leeward side. This should, however, not mean that the Smith & Barstad model is bad, and it was definitely not bad at the time when it was developed.

**We introduced an example that compares our model to the Smith & Barstad model in order to point out the fundamental differences in predictive power more clearly (lines 427–485 and Fig. 5) and an application to a real-world topography (Sect. 7), but also point out that tuning the model parameters towards a given precipitation pattern is not necessarily a good validation for models of this type.**

*If the focus is the latter, I recommend authors to revise the title and importantly compare their results with earlier co-evolution studies. Many co-evolution modeling studies have been published recently, and most of them are not mentioned in this manuscript. I suggest authors first to check the following paper, just published in HESS, and references cited in that paper.*

*Paik, K. and Kim, W.: Simulating the evolution of the topography-climate coupled system, Hydrol. Earth Syst. Sci., 25, 24592474, https://doi.org/10.5194/hess-25-2459-2021, 2021.*

*Even the focus is on co-evolution itself, some validation of a new orographic model is still desired. However, as long as the authors make good scientific contributions with their modeling, the necessity for thorough validation is less important than the previous story. There have been some earlier studies that I remember which adopted very simple orographic models with little validation but were published due to their independent scientific contributions.*

*Below, I provide more technical comments.*

*L39: This concept, i.e., in reality the flow discharge, instead of the drainage area, controls the erosion is not new. For example, it was stated in Paik (2012 ref below) as "While the above equation expresses the erosion rate as a function of the drainage area, it should be the flow that contributes to the bedrock erosion in reality. In the formulation of empirical equations, the drainage area has often been chosen as a surrogate of the flow discharge due to the difficulty of measuring flow discharge. However, there is no need to use the drainage area instead of flow in the numerical modeling."*

*Paik, K.: Simulation of landscape evolution using a global flow path search method, Environmental Modelling & Software, 33, 35-47, https://doi.org/10.1016/j.envsoft.2012.01.005, 2012.*

*Some notations are not defined in the text, e.g., $u_{v/c}$ in equation (3).*

Our list of studies on co-evolution was indeed not exhaustive, and we are really sorry that we missed your very recent work in HESS. **We studied the reference list of your recent paper and added the references we found important in this context (Anders et al. 2008, Colberg & Anders 2014, Han et al. 2015, Paik & Kim 2021) (line 19).**

It was not our intention to claim that the idea was new. At least the studies cited in line 38 of our manuscript used this concept. The models can, of course, easily be written in terms of discharge. However, many concepts in the field of landscape evolution and tectonic geomorphology (e.g., erodibility, steepness index) refer to catchment size. Therefore, the models usually involve an actual precipitation and a reference precipitation, while we prefer the concept of the catchment-size equivalent described at the end of the paragraph (Eq. 2). To be frank, we have no idea what to do with the comment.

We thought that defining $u_v$ and $u_c$ and speaking of the "respective" property, it should be clear that $u_{v/c}$ can be either $u_v$ or $u_c$. **Anyway, it is mentioned explicitly now (lines 118–119).** And since we did not find "some notations" that are not defined in the text, we would be happy to receive some help in finding them.

*L140: I personally had also been tempted to use this approach. But I have had the following peer comment on this idea some time ago: ". . . the temperature change with mean elevation change is not likely to represent a reasonable assumption. Yes, the atmosphere is cooler over mountains – but the air source for the precipitation is over an ocean and topography is not going to force the air to be cooler upwind of it." I still advocate this equation but you would need some supporting argument for its use.*

Maybe we missed the key point of your colleague's comment. The decrease of temperature with increasing altitude is the basis of all models in this context. If the air is blown uphill, it is of course cooled down already by adiabatic expansion. In principle, this is also the basis of the Smith & Barstad model that you used in your study. So we are confused that someone convinced you that your approach was not good and you moved to an existing model that makes basically the same assumption. Or does your argument really refer to the upwind direction, which would be towards the ocean? Of course, the air above the ocean is not cooled down by mountains on land, but this is assumed in none of the models. So we are really confused by the statement of your colleague.

*L364-365: If this is correct, it can be a strong reason to develop an alternative model. But you should demonstrate it in comparison with real observation and Smith and Barstad (2004) model to convince it.*

Of course, it is correct that the Smith and Barstad (2004) cannot capture large-scale patterns. Smith & Barstad already mentioned the respective scales in their paper (Eq. 7), and the modified source term introduced in their Section 3 ("airflow dynamics") does not change this behavior fundamentally. These scales are some tens of kilometers at maximum, while precipiation typically decreases exponentially with distance from the ocean at a scale of several hundred kilometers, which cannot be captured by the Smith & Barstad model. This was one of the reasons why we stopped trying to extend the Smith & Barstad model gradually and decided to design a new model from the scratch. **We intoduced a more detailed comparison of our model to the Smith & Barstad model using a ramp-plateau topography in order to show more clearly which phenomena cannot be captured by the Smith & Barstad model (lines 427–485 and Fig. 5).**

*Section 7.1: The simulation domain here is only about a few hundred km. How can these simulations capture the 'continentality'?*

At least in unforested areas, the decrease in precipitation with increasing distance from the ocean seems to be quite strong. Makarieva et al. (2009, doi 10.1016/j.ecocom.2008.11.004) found an exponential decay with a length scale of 600 km. As discussed in Sect. 8.1, the value $L_l = 600$ km corresponds to a decay scale of 1200 km at 50 % evapotranspiration (Fig. 8a). So the scales used in Sect. 8.1 are quite well-suited for illustrating under which conditions even the effect of "continentality" alone affects landform evolution.

**Reviewer 2 (Sebastian Mutz)**

*Assessment of the study's contribution*

*The manuscript represents a potentially very important contribution to model-based approaches in the field of tectonics-landscape-climate interactions. A common problem in landscape evolution modelling is the efficient inclusion of realistic orographic precipitation, since General Circulation Models (GCMs) have weaknesses in representing such precipitation, and non-hydrostatic regional climate models (RCMs), which are able to represent orographic precipitation much better, are equally complex and also have high computational requirements. An efficient orographic precipitation model, that is able to respond quickly to orographic changes produced by landscape evolution models (LEMs) or prescribed topography therefore bridges this gap in modelling. The authors address an important and widely recognised gap by presenting an alternative to the previous, simple orographic precipitation models, such as Smith and Barstad's model based on linear theory for orographic precipitation (LTOP). While the LTOP model has increased in complexity over time and represents a viable option for LEMs, the model presented here has some advantages over it, and model diversity in general increases the overall reliability and knowledge gain of the community's modelling efforts. The presented study therefore is, in my eyes, a very valuable contribution to the LEM and Earth system science community in particular.*

*General Comments*

*The manuscript is well written and generally easy to follow, as is appropriate for a manuscript that is of potential interest to different geoscientific communities. The authors present the readers with backgrounds on SPIMs, the need for simple orographic precipitation models, and how the model presented here complements previous approaches. I believe this is appropriate given the contribution assessment above. The readers are talked through the governing equations and model in sufficient detail to develop a feeling for the model's potential applications and limitations. The demonstrations (section 7) are particularly useful for the LEM community. The conclusions are helpful for readers to determine the suitability of the model for their purposes. I do, however, have a few (mostly minor) concerns about this study. I believe these can be addressed fairly easily:*

*1. Title: Since the focus of this study – judging by introduction, examples and references – currently lies on presenting an orographic precipitation model specifically for LEM/geomorphology community, I think it is better for the title to reflect that when it is published in a journal that also sees publications of climate models "for climatologists". If the manuscript is intended to simply present an orographic precipitation model, the text would have to be adjusted to highlight how it fits into the realm of climatology/meteorology and its vast model landscape. Given that this type of model is likely most needed in the geomorphology/LEM community, I would simply adjust the title here rather than change focus of the manuscript.*

You are right – it was not clear from the title that the target community is rather landform evolution modeling than climatology. **We adjusted the title accordingly (including the definition of a model name and acronym as suggested by one of the executive editors).**

*2. The study's focus (only a potential concern): If the study's focus is the perceived one (described above), my only concern is the title. The model of course has potential applications beyond the geomorphological community. However, if the idea is to address a wider audience in this particular manuscript, I would expect much more discussion of its fit into the climate model landscape, as well as (performance and skill) comparisons to models that are well established in climatology for precipitation simulations in orogens (e.g. WRF), for example by application of the model presented here to a region already investigated with WRF and/or other models (ideally of varying complexity).*

*3. Model validation: The manuscript describes well the conceptual differences between this and comparable models (e.g. LTOP), and the model construction seems very reasonable. However, it is not clear what its prediction skill is compared to other models. There is no application of the presented model to a real setting, followed by a comparison to observational data or other comparable models. Esp. for scientists interested in applying the model outside a purely theoretical framework, this lack of validation is problematic and should be addressed.*

*4. The manuscript lacks discussion of the potential applications (and caveats) of the model outside the more theoretical realm/sensitivity experiments. The point above is one way to address this. Furthermore, I imagine that this model is of great interest to those investigating the co-evolution of orogens, climate and landscapes for real settings and times in the past. To do that, however, a number of additional steps need to be taken (see specific comment for L48-51). I think a discussion of this would increase this study's usefulness and also avoid ill-informed use of the presented model.*

Since the climatology community is probably larger than the landform evolution modeling community, it would be tempting to address a wider audience. This would, however, require a very thorough analysis of the two simplifications, (i) the limitation to a pre-defined atmospheric flow field (in this version even uniform) and (ii) the simplified consideration of the conversion of moisture and precipitation. We feel that this would we rather a separate study of the "model comparison paper" type, and we would not be able to do this on our own without a partner from the climatology community.

We agree that the prediction skills should be discussed more thoroughly. **So we added two considerations: (i) An example that compares our model to the Smith & Barstad model in order to point out the fundamental differences in predictive power more clearly (lines 427–485 and Fig. 5). (ii) An application to a real-world topography (Sect. 7). However, we also feel the need to point out that tuning the model parameters towards a given precipitation pattern is not necessarily a good validation for models of this type.**

Indeed a good point! **We added a section "Scope, limitations, and perspectives" (Sect. 9) about these aspects.**

*5. Equations (minor point): Each term in the equations, starting in the introduction or at least from the very beginning of section 2.1, should be given units explicitly. Partially, this suggestion may stem from the way I think of and follow/read equations (I find it more difficult to think them through without units in front of me), but it would enhance reproducibility and help avoid confusion regardless. I strongly suggest clearly stating the units for each of the terms in the equations throughout the entire manuscript, even if they are just the SI units the terms are usually expressed in. I also recommend going through all again carefully to catch possible oversights during write-up (see specific comments).*

Of course, the units are essential for recapitulating equations and also serve as a first check whether the there is something wrong. However, for us it is rather taking a sheet of paper and rewrite the equations, where each variable is replaced by its unit, and then canceling identical units. Writing units directly into the printed equations (e.g., in brackets) makes the equations cumbersome and only makes sense in rare situations where non-integer exponents occur, but this is not the case here. In general, we prefer to choose symbols in such a way that it is easy to remember the units, e.g., all $L_{...}$ for horizontal lengths, $H_{...}$ for heights, or greek letters for nondimensional properties. **We added remarks on the unit at a few places (lines 118, 126, 146, 148, 163–164, 166, and 241), but not in the displayed equations.**

*6. Code documentation: This point is not directly related to the manuscript, but important for potential users nevertheless. As someone who is actually interested in applying this model, I downloaded the code for openLEM from the link provided here. My go-to language for modelling (and most other things) is Fortran, but I usually don't have issues following C++ code if it is well commented and/or documented. However, it is difficult to locate the relevant code if I am interested in only the orographic precipitation model (decoupled from openLEM). Much of it seems to be in orogen.cpp, but much of the code lacks sufficient comments to navigate easily. I think a clean documentation, more comments and orographic precpitation model packaged as a separate model (decoupled from openLEM) will remove barriers for other scientists to use it. I appreciate the explicit offer of assistance in the "code and data availability" statement, but think a an independence of the authors' assistance through documentation benefits everyone, including the authors.*

In fact, it was the code openlemprecip11.cpp in the repository, but now it is included in the latest OpenLEM version. **Anyway, we now also provide a commented standalone version that can easily be translated to other programming languages on the OpenLEM homepage (http://hergarten.at/openlem/lfpm.php).**

*Technical/Specific Comments*

*Below, I suggest a few small corrections that came to mind during reading.*

*L4: GCM coupling not only increases the complexity, but GCMs also have notable weaknesses in representing precipitation, esp. in mountanous regions. That is arguably the bigger problem of using GCMs. In case of RCMs like WRF, "only" the increased complexity and high computational demands remain a problem. I suggest a small adjustment to this statement in the abstract.*

Right – **we adjusted it to "regional climate models" in the abstract (line 4).**

*L18: ". . . the the geometry . . . ", omit one "the"*

Thanks for finding this mistake – the wording was changed anyway here since we added references which did not fit into the scheme perfectly.

*L29: Maybe change to ". . . all particles are immediately excavated once detached from bedrock." for better readability.*

Following the suggestions of a reviewer of another manuscript, **we have modified the entire description of the SPIM and its derivates slightly (lines 28–36).**

*L48-51: In addition, GCMs would not be suitable tools for predicting orographic precipitation [e.g. Meehl et al. 2007], esp. not at the catchment scale (see above comment). However, RCMs come with the same computational drawbacks the authors mention here. I suggest highlighting this point here. That said, once a study is upscaled for larger orogens in studies of how their evolution is linked to climate, landscape evolution and erosion, the changes in large scale surface uplift has significant impacts on regional and global climate [e.g. Takahashi and Battisti, 2007; Paeth et al., 2019], and thus on the boundary conditions (moisture availability, wind and therefore advection velocity , etc.) for RCMs or less complex orographic precipitation models like LTOP or the model presented here. This means that once larger changes are introduced to an orogen, there is no way around running GCMs, even if they then simply drive simpler orographic precipitation models rather than RCMs. The same is true once we leave the realm of sensitivity experiments and look at an orogen in the geologic past, when palaeoenvironmental boundary conditions create a very different global climate and thus change the input fields for any RCM or simple orographic precipitation models [e.g. Mutz and Ehlers, 2019]. The need for GCMs for such upscaled experiments ought to be highlighted somewhere – here or (probably more fittingly) in a "caveats/warning" paragraph in conclusions, or both.*

**We adjusted the text here towards RCMs instead of GCMs and discussed the problem of the output a bit more thoroughly (lines 53–62). In order to address the potential caveats, we added a section "Scope, limitations, and perspectives" (Sect. 9).**

*L79: I would describe it more accurately as "the goal of this study"; the goal of the paper is to present the study/model.*

Indeed – **we adjusted it (line 100).**

*L97: I suggest giving discharge a different symbol in the introduction to avoid potential confusion altogether. If the authors think it is merited to reference $q$ in context of discharge anyway, this may be done by adding a side note a la "[new symbol] is discharge, often denoted as $q$ in other manuscripts, . . . " in the introduction.*

If the capital $Q$ was not occupied by the sediment flux in the examples section, this would have been a good choice. Using a completely different symbol would not be very intuitive, and since the atmospheric fluxes introduced later are $q_v$ and $q_c$, it should not be a big source of confusion. **Anyway, we pointed out that $q_v$ and $q_c$ have a meaning different from $q$ without subscript (line 121).**

L135-144: Equation 15 does not follow 14 as it currently stands. However, the flaw seems to be in 14. If $\beta/\beta_0 = e^{-[a/(T_0-\Gamma H)]}/e^{-[a/T_0]}$, then 14 should read as $e^{-[a/(T_0-\Gamma H)-a/T_0]}$, i.e. the last term in the exponential should be subtracted if I'm not mistaken. 15 would then follow 14 again, so I think it's simply a matter of getting a sign wrong during the write-up of the manuscript. For 16, it's not clear from the text why the $-T_0\Gamma H$ term in the denominator is considered negligible.

You are not mistaken, the way you wrote it is definitely correct. However, you just wrote $e^{-(A-B)}$, while our Eq. 14 $e^{-A+B}$, so the same. **We added an explanation under which conditions the term $-T_0\Gamma H$ can be neglected (lines 179–181).**

L534 (Fig.9): The coloured dots next to uplift rates are somewhat difficult to make out. Furthermore, I suggest adjusting colours to take into consideration common forms of colour blindness. This is a general recommendation, but something I notice every time I see red next to green as here. If that has been considered when these particular shades were picked, please ignore my second comment.

We agree that the colors were far from ideal for color-blind people. **We have changed the colors following rules for coloring for colorblindness in a way that the figure should now also work for people with protanopia, deuteranopia, or tritanopia. Furthermore, we removed the tiny dots next to uplift rates and instead changed the color of annotation accordingly.**

L181 (Fig. 10): I suggest changing the colour scale to something other than the rainbow colours (e.g. a simple grey scale) (1) to make visualisation more accessible (consider colour blindness), and (2) because the rainbow scale has been demonstrated to be misleading due to the lack of clear perceptual ordering.

We agree that rainbow colours are problematic for colorblind people. However, despite reading suggestion for coloring plots for colorblindness and experimenting a lot, we were not able to achieve a satisfactory solution where the topography in the background and the color-coded drainage network on top of it still show the characteristic properties we are describing next to the figure. In order to be as inclusive as possible, but at the same time provide the best possible visual representation of our results for people without this limitation, we offer a separate version of this figure for colorblind people in the supplement. There we have removed the background (topography) and color-coded the drainage networks with a cubehelix color palette, which should work for people with protanopia, deuteranopia, or tritanopia.

I hope my input here helps polish the manuscript somewhat and look forward to seeing a revised version.

---

## Referee Report (RR1)

**General Comments**

The authors' revisions represent a significant amount of work and address all of the major shortcomings of the previous version of the manuscript. The authors have addressed all important points I made and concerns I had well. Most notably, the focus of this manuscript is now clearer than before thanks to changes in text and title. (I think the change in title now better reflects the focus of the study and will catch the eyes especially the geomorphology and LEM community, which will likely benefit from this work the most.) Moreover, the authors addressed the noted lack of example-based comparisons to similar models. This is now addressed also by example presented in the new Fig. 5 – this is an important addition in my opinion, since it demonstrates by example some key differences between the LFPM and the commonly used SB model. Furthermore, the discussion was enhanced with a section on limitations and perspectives, which I believe will enhance the study's usefulness (and ensures a responsible and more informed use of the LFPM model).

**Technical/Specific Comments and Suggestions**

Line 95: Change to "It was adopted [...]"

Line 171: This seems identical to eq 14-15 in the previous version of the manuscript (see my previous comments on L135-144 in the first review) except written on one line. Please double check.

Fig 5: Add the dashed line (ramp "topography" to the legend).

Best wishes
Sebastian Mutz

---

## Author Response (AR2)

Dear Travis O'Brien,

thank you very much for taking over the editorial handling and for your encouraging comments! Please find below our responses, where changes to the manuscript are highlighted in bold letters. Line numbers refer to the version with highlighted changes.

Best regards,

Stefan Hergarten und Jörg Robl

**Editor (Travis A. O'Brien)**

**Technical issues**

*1. The model name should include a version number or other unique identifier in the title (e.g., LFPM 1.0): see https://www.geoscientific-model-development.net/about/manuscript_types.html for the policy on titles in "Model description papers"*

**Fixed**.

*2. I identified some grammar issues and typos that should be fixed:*

*line 64: "approaches were" → "approaches have been"*

We were not sure here initially; **fixed (line 64).**

*line 75: "in upwind" → "in the upwind"*

**Fixed (line 75).**

*line 90: "I was adopted" → "It was adopted"*

**Fixed (line 89).**

*line 703: "respectiv" → "respective"*

**Fixed (line 721).**

**Additional comments**
**Symbol choices:**

*The symbol choices made this section somewhat hard for me to follow, since the convention in climate/atmospheric science is that $q$ is atmospheric moisture, and $u$ and $v$ are the E/W and N/S wind components. You might consider using $Q$ for the column integrated water, $U$ for the column integrated wind, and $F$ for the horizontal moisture flux.*

Makes sense, **so we changed $u \to Q$ and $q \to F$.** However, we kept the lowercase $v$ for the velocity since our x-axis is not necessarily N-S or E-W oriented, and $U$ would overlap with the uplift rate in the examples of landform evolution.

*It might also be worth stating that this equation can be derived directly from a column-averaged moisture budget (with the exception of the dispersion term, which I have a comment on below).*

Yes, it is nothing but the column-integrated moisture budget. **We added it in line 118.**

*It might also help declutter the equations to annotate variables in Equation 4 as something like $Q_i$, with $i \in [v, c]$.*

After thinking about it, we arrived at the point that it may be good for the moment for decluttering the equations, but the notation v/c may be clearer when it occurs again later. And when the equations become more complicated later in the paper, it would not help. So we kept it.

**Consider simplifying to only one advective velocity:**

On line 109, $v_{v/c}$ is introduced as the advective velocities for water vapor and water condensate. While it physically is correct that the two need not have the same vertically-averaged advective velocity (since this advective velocity can be interpreted as the moisture-weighted, vertically averaged wind velocity), it turns out not to matter. The only place where they appear later on is in the definition of $\beta$, which ends up being associated with a tunable equation. If instead $v_v = v_c$, then $\beta = \alpha$, and all the arguments for the form of $\beta$ still hold. Therefore, it may help to declutter the equations if the subscript is omitted (perhaps with a comment justifying the choice to do so).

Right, we thought about this when writing the first draft. However, we feel that it does not make much difference whether we start from different velocities and show in a few steps that it has no effect or whether we justify the choice of a single velocity. Finally, we found it safer for the review process to show that it makes no difference.

**The dispersion term:**

I was initially somewhat baffled by the motivation for the form of the dispersion term at first. If one uses Reynolds averaging on the column-averaged moisture budget equation, with $F \equiv \overline{Q} \cdot \overline{U}$ (using my notation above, and where the overline represents a spatial average operator), then a second order term appears that represents the subgrid transport of water: $\nabla \cdot (\overline{U'Q'})$. The form of the dispersion term in your Equation 4 implies that the subgrid flux of water is equal to

$$\overline{U'Q'} = L_D \cdot \frac{\partial \overline{QU}}{\partial y}$$

The above implies that the subgrid flux operates only in the y-direction and that its magnitude is proportional to the gradient of the flux in the y-direction. This can roughly be interpreted as "horizontal shear in the flow leads to cross-shear transport of water." This seems like a reasonable approximation to the effect that subgrid eddies– which would be generated by horizontal shear– might have on the transport. It might be worth elaborating this in the manuscript, since otherwise it is difficult to understand what is the physical motivation for form of the dispersion.

The description and motivation of the dispersion term was indeed a bit rushed, perhaps we are familiar with this concept in other fields, but not so much in the context of turbulent flow. **We tried to improve it (lines 123-137).**

**Reviewer 2 (Sebastian Mutz)**

Line 95: Change to "It was adopted [...]"

**Fixed (line 89).**

*Line 171: This seems identical to eq 14-15 in the previous version of the manuscript (see my previous comments on L135-144 in the first review) except written on one line. Please double check.*

Yes, indeed the same since it was already correct in the first version. **Anyway, we have now introduced one more step and wrote it in the form $e^{-(x-y)}$ instead of $e^{-x+y}$ in order to avoid any confusion (line 173).**

*Fig 5: Add the dashed line (ramp "topography" to the legend).*

**Done.**